

# Stratification and Mixed Layer Depth around Iceland, characterization and inter-annual variability

Angel Ruiz-Angulo [1*], Esther Portela [2], Charly de Marez[1], Andreas Macrander[3],

Sólveig Rósa Ólafsdóttir[3], Thomas Meunier [4], Steingrímur Jónsson [3,5], and M. Dolores

Pérez-Hernández [6]

[1]Earth Science Institute, University of Iceland, 101 Reykjavik, Iceland

[2]Univ. Brest, Laboratoire d'Océanographie Physique et Spatiale, CNRS, IRD, Ifremer, Plouzané, France

[3]Hafrannsóknastofnun / Marine and Freshwater Research Institute, Hafnarfjörður, Iceland,

[4]Woods Hole Oceanographic Institution, Woods Hole MA, USA

[5]University of Akureyri, Akureyri, Iceland

[6]Unidad océano y clima, Instituto de Oceanografía y Cambio Global, IOCAG, Universidad de Las Palmas de Gran Canaria, ULPGC, Unidad Asociada ULPGC-CSIC, Las Palmas de Gran Canaria, Spain

*Correspondence to*:  Angel Ruiz-Angulo (angel@hi.is)

**ABSTRACT**

The ocean around Iceland witnesses some of the most important transformations of water masses that drive the global ocean circulation. Here, we analyze 29 years (1990-2019) of quarterly hydrographic sections data collected around Iceland. The hydrographic properties around Iceland show important spatial variability. Based on temperature, salinity, and stratification structure, we classified the Icelandic waters in three distinct regions: the South, the North and Northeast regions. The warm and salty Atlantic Waters that dominate the south show the deepest winter mixed layers (~500m) while the North and Northeast show shallower depths (~100m). Based on the decomposition of total stratification into temperature and salinity contributions, we find that, in the South, the subsurface stratification is mainly dominated by temperature, in the Northwest salinity dominates, while in the North, the seasonality of the North Icelandic Irminger Current and East Icelandic Current alternate the temperature and salinity contribution to stratification. The interannual variability of the mixed layer and of its thermohaline properties is also large around Iceland. Mixed layer waters were generally colder in the 90's, then warmed until approximately 2015, and became colder again from 2015 to 2018. In the Northeast, a clear multidecadal mixed layer warming trend clearly emerges from the interannual variability as the Atlantic Water progresses northeastward, which is responsible for transforming locally, the upper stratification from salinity dominated into temperature dominated, allowing for the formation of deeper mixed layers. This is associated with the "Atlantification" of the Arctic. Elsewhere, we observe density-compensated changes in mixed layer temperature and salinity, without clear trends. This study provides an unprecedented and detailed description of the seasonal to multi-decadal variability of the mixed layer depth and stratification around Iceland, and their link with the changing North Atlantic under global warming.

**Keywords:** Mixed layer depth, Mixed layer properties, stratification, Ocean warming, Atlantification, Interannual variability



## 1 INTRODUCTION

The Nordic Seas, together with the Irminger Sea and the Iceland Basin, play a crucial role on the Atlantic Meridional Overturning Circulation (AMOC). The Nordic Seas are among the few places on the globe where the formation of deep waters (1000-3000 m depth) occurs during winter deep convection (Petit et al., 2020). The southern end of the Nordic Seas is bounded by the Greenland-Iceland-Scotland ridge (GISR). The North Atlantic Current (NAC) brings the warm and salty Atlantic Water (AW) northward into the Nordic Seas (Hátún and Chafik, 2018; Østerhus et al., 2019; Hátún et al., 2021). The AW crosses the ridge in three ways (Fig. 1): (i) between Greenland and Iceland, where the Irminger Current (IC) forms the North Icelandic Irminger Current (NIIC) bringing AW flowing clockwise around Iceland (Jónsson & Briem, 2003; Jónsson & Valdimarsson, 2012). (ii) Between Iceland and Faroe (Mauritzen, 1996), and (iii) through the Faroe Shetland Channel (Hansen and Østerhus, 2000; Hansen et al., 2023), contributing with up to 48% of the total AW transport. The AW undergoes strong cooling and densification in the Nordic Seas and the Arctic Ocean (Våge et al., 2015, 2018; Pérez-Hernández et al., 2019; Athanase et al., 2020). This modified AW is referred to as Atlantic-origin Overflow Water (AtOW; *e.g.,* Havik et al., 2017; Casanova-Masjoan et al., 2020) and is one of the two sources of Denmark Strait Overflow Water (DSOW; Semper et al. 2019). AtOW travels southward as a mid-depth water mass in the East Greenland Current (Håvik et al., 2017), from where, part of it diverts east and merges with the NIIC northeast of Iceland (Casanova-Masjoan et al., 2020).

The transformation of AW into AtOW takes place in different areas of the Nordic Seas: along the Norwegian Current western intrusions (Håvik et al., 2017), in the Iceland Sea Gyre (Våge et al., 2013), on the eastern side of Greenland, or even -due to its proximity- in the Arctic Basin (Pérez-Hernández et al., 2019). This transformation has different driving mechanisms impacting mixing and convective processes. Wind-stress and sea-ice retreat drives transformation east of Greenland (Våge et al., 2018), sea-ice retreat and heat exchange dominate north of Svalbard (Pérez-Hernández et al., 2019; Athanase et al., 2020), and heat fluxes are the main driver on the center of Iceland Sea (Våge et al., 2013). Thus, the Nordic Seas have been previously described as a "mixing pot" (Renfrew et al., 2019), largely responsible for the overall formation of deep overflow water (Lozier et al., 2019). The Nordic Seas are also a large repository of freshwater arising from glacier/river discharge and from the Arctic. This water mass increases buoyancy and is carried southward by the East Greenland Current (EGC). Therefore, it is crucial to fully understand the variability of the upper ocean, where mixed layers (ML) develop and transform these water masses.

The Arctic Ocean is warming much faster than the global average, a process known as the "Arctic Amplification", which is also associated with the "Atlantification" of the Arctic (Polyakov et al., 2017; Dai et al., 2019). Even though the causes are still in debate, there are evident observed consequences of the Arctic Amplification, such as the decrease in the extent of seasonal sea-ice extent (Dai et al., 2019) and a weakening of the cold halocline (Polyakov et al., 2020). Changes in temperature and salinity in the upper ocean modify the upper ocean stratification, which partially controls the mixed layer depth (MLD).

The depth and structure of the ML is primarily controlled by local buoyancy forcing, i.e., surface heat loss and freshwater fluxes, which modifies the water density (Kohler et al., 2018). Wind forcing contributes by enhancing turbulent mixing, deepening the ML under certain conditions (Petit et al., 2020). For instance, within the Iceland Basin, wintertime buoyancy loss drives deep convection, shaping the thermohaline properties that influence the lower limb of the AMOC and its variability in the subpolar North Atlantic (Petit et al., 2021). The pre-existing stratification of the water column is responsible to control the effect of the surface forcings. Strongly stratified upper layers resist mixing while weak stratification allows deeper penetration of turbulence and convection mixing (Pierce et al., 1986). Over shorter timescales, on the order of days, the MLD can significantly deepen as a result of the strong wind events with significant wind stress





and associated large wave heights (Skyllingstad et al., 2023).

The IPCC report indicates with *high confidence* that roughly the 40% of the global ocean mean upper ocean stratification has increased about 3.3–6.1% since 1960 due to both ocean warming and high latitude freshening (Tesdal et al., 2018; Yamaguchi and Suga, 2019; Bindoff et al., 2019; Liu et al., 2020; Salleé et al., 2021). Increased stratification is associated with less efficient mixing, reducing the exchanges of heat and tracers from the mixed layer into the ocean interior. It has also been observed, with *high confidence*, that the ML is undergoing changes (Bindoff et al., 2019; on Climate Change, IPCC). Particularly, the shallow summertime ML, which is more likely to be affected by global warming, is deepening at

a rate of $5 - 10$ m per decade (Sallee et al., 2021). Despite the reported global patterns, it has been also acknowledged that regional changes might differ from the global estimates (on Climate Change, IPCC).

The warming of the ML and the associated increase in stratification influence biogeochemical processes like phytoplankton blooms and carbon or oxygen sequestration, key components for the Earth's climate (Olafsson, 2003; Pérez et al., 2021). In the waters surrounding Iceland, the phytoplankton community is closely linked with the water mass

properties and hence, an "Atlantification" will replace polar communities with more Atlantic communities (Cerfonteyn et al. 2023). In the Arctic, north and northwest of Iceland, the early onset of stratification in spring gives rise to rapid shallowing of the mixed layer and triggers early spring phytoplankton blooms, whereas the weakly stratified water-column in the Atlantic water and associated deep ML delays the spring bloom south of Iceland (Zhai et al., 2012). This also has strong consequences in carbon uptake, vertical nutrient supply and biological processes (Yamaguchi and Suga, 2019).

Other indirect impacts of the increased stratification include changes in upwelling, deep-water formation rates, biological production, and remineralization rates (Holt et al., 2016), and deoxygenation (Shepherd et al., 2017).

At the regional scale, it is challenging to determine the extent to which changes in stratification and in water-mass properties are driven by natural or human-induced variability. Moreover, the relatively short observation records in most of the oceanic regions hinders the attribution of the observed changes to the driving mechanisms and forcings. In this study,

by using a long time series of hydrographic observations around Iceland spanning 29 years, we aim at unraveling the different scales of variability of the water-mass properties, MLD and stratification around Iceland.

## 2 DATA AND METHODS

We use Conductivity-Temperature-Depth (CTD) data from the repeated hydrographic observational program of the

Icelandic Marine and Freshwater Research Institute between 1990 and 2019. The oceanographic surveys took place quarterly, mainly in February, May, August, and November with little coverage during the intermediate months. Observations are made at standard repeated sections. The profiles are obtained with a Seabird 911plus CTD mounted on a rosette with Niskine bottles. The conductivity data are calibrated with salinity samples taken at the bottom of each station. All sensors underwent regular calibrations by the manufacturer.

In our analyses we considered only the deepest stations in each section (red dots in Fig. 1), including nearby stations within an area defined by $1° × 0.5°$ in longitude and latitude (red boxes). The selected stations are located outside of the Icelandic shelf (about 500 m depth). This criterion was chosen to avoid topographic effects, such as across shelf processes on the stratification of the water column and to avoid MLDs limited by shallow bathymetry. Thus, the stations in gray, HB and IB in Figure 1 were not considered as they fall on the shelf. For the sake of simplicity these stations will be named





with the acronym of the standard section, first two letters and the station number. The station full name (section and station number) can be found in Table 1.

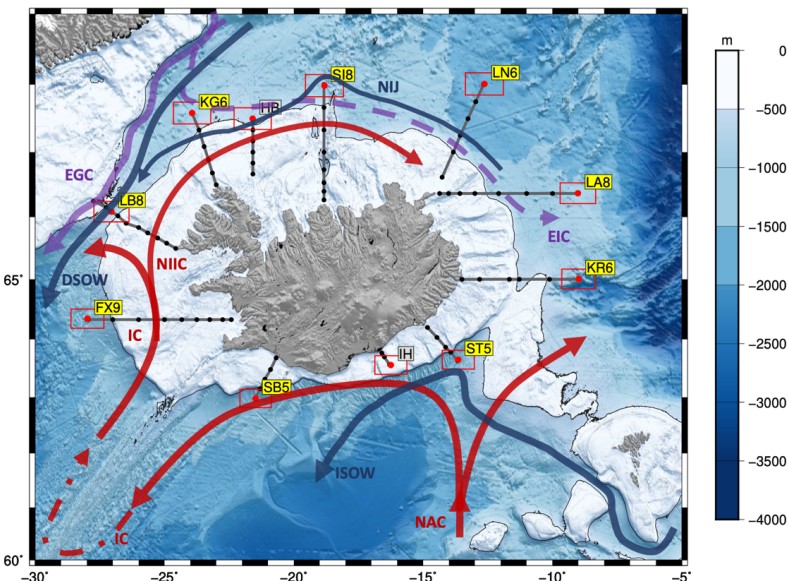

**Figure 1: Map of the typical hydrographic sections collected by the Marine and Freshwater Research Institute around Iceland; the black dots represent the nominal location of the standard stations from 1990-2019. The red dots are the stations used for this analysis**
**and the red boxes delimit the area within which all data were considered for this study. The grey bathymetric contours are spaced every 100 m for the shallow water until the 500 m depth (thick black line) and then every 500 m. The hydrographic stations shown in the yellow boxes corresponding to the standard sections: Faxaflói (FX9), Látrabjarg (LB8), Kögur (KG6), Hornbanki (HB), Siglunes (SI8), Langanes NE (LN6), Langanes E (LA8), Krossanes (KR6), Stokksnes (ST5), Ingólfshöfði (IH) and Selvogsbanki (SB5). The gray labeled IH and HB were not used in this analysis. The main surface and deep currents are also depicted on the map.**


In this study we analyze the inter-annual variability and linear trends of the ML over a 28-year period as well as the seasonal variability using the seasonal extremes (summer and winter), when there is more data coverage. From the CTD stations we estimated the MLD using the density threshold method with a criterion of $\sigma\theta = 0.01$ kg m$^{-3}$ (as, for instance, in Piron et al., (2016) in the Irminger Sea) and a reference depth of 10 m. We chose this criterion instead of the usual 0.03 kg
m$^{-3}$ (De Boyer Montegut, 2004) as the latter overestimated the MLD in more than 500 visually inspected profiles (not shown). For comparison and robustness of our chosen method, we also estimated the MLD using other criteria (de Boyer Montegut et al., 2004; Holte et al., 2017). However, we found the density threshold method appropriate for our region as it had a great score even for cases where the variations of salinity and temperature were large. Those variations usually compensate in density making this method more suitable.

For each profile we computed the Brunt -Väisälä frequency ($N^2$), defined as:

$$N^2 = g\frac{1}{\sigma\theta}\frac{\partial\sigma\theta}{\partial z}. \tag{1}$$

Where g is the gravity acceleration, $\sigma\theta$ is the potential density and z is depth. $N^2$ can be decomposed on the relative contribution of the salinity and temperature to the observed stratification as follows:





$$N^2 = N_T^2 + N_s^2. \tag{2}$$

Where $N_T^2$ and $N_s^2$ are the components representing the stratification set by the temperature and salinity respectively and are defined as:

$$N_T^2 = g\left(\alpha \frac{\partial T}{\partial z}\right), \tag{3}$$

$$N_s^2 = g\left(\beta \frac{\partial S}{\partial z}\right). \tag{4}$$

Where $\alpha$ is the thermal expansion coefficient and $\beta$ is the haline contraction coefficient at constant pressure. This decomposition has also been made to classify the oceans by their stratification contribution into $\alpha$-ocean, $\beta$-ocean, and transition zone, where in $\alpha$-oceans stratification is permanently dominated by temperature, in $\beta$-oceans by salinity and the transition regions are either intermittently or seasonally dominated by temperature or salinity (Carmack, 2007; Stewart and

Haine, 2016). For the water column to be statically stable, $N^2$ must be positive. However, the contributions may not be positive; when any of its components, $N_T^2$ or $N_s^2$ are negative, temperature or salinity respectively have a destabilizing effect on the resulting stratification that must be compensated by the other variable to maintain a stable water column. Small values of $N^2$ indicate that the water column is weakly stratified which favors mixing due to winter convection and deeper MLD.

To investigate furthermore the driving mechanism of the MLD we used a one-dimensional model (Price et al., 1986) initialized with ERA-5 wind stress, heat, and freshwater fluxes (Hersbach et al., 2020) and the winter averaged vertical profiles of temperature and salinity from the observations presented here. The 1D model would reveal the contribution from diurnal heating/cooling freshwater fluxes and wind mixing. In addition, we broaden the impact of our findings by using the hydrographic database published in Brakstad (2023) that includes, in addition to the dataset from the Marine and

Freshwater Research Institute of Iceland, other multiplatform observations like Argo floats or cruise data between 1950 and 2019. For this objective, a larger oceanic region is used and classified into $\alpha$-ocean and $\beta$-ocean using the spice frequency, $K^2$, (Carmack, 2007; Strehl et al., 2024), defined as:

$$K^2 = N_T^2 - N_s^2. \tag{5}$$

$K^2$ is positive (negative) in $\alpha$-($\beta$-)oceans.


### 3 RESULTS

#### 3.1 Hydrographic properties around Iceland

The hydrographic properties (potential temperature-salinity diagrams) around Iceland show important spatial, seasonal and interannual variability are shown in Figure 2; the T/S properties differ widely between the three representative stations:

FX9, SI8 and LN6 for the west, north and northeast of Iceland. FX9, in the southwest of Iceland is completely dominated by Atlantic Water (AW; Fig. 2 a and b). At SI8, in the North, the dominating water masses in winter are of polar origin, i.e., warm Polar Surface Water (PSWw) in the upper layers, and Atlantic Overflow Water (AtOW) and Arctic Overflow Water (ArOW) in the intermediate/bottom waters (Fig. 2c and d). The SI8 station also presents the largest dispersions of its thermohaline characteristics. LN6, in the Northeast, contains the coldest and densest waters on average (Fig. 2c and f).




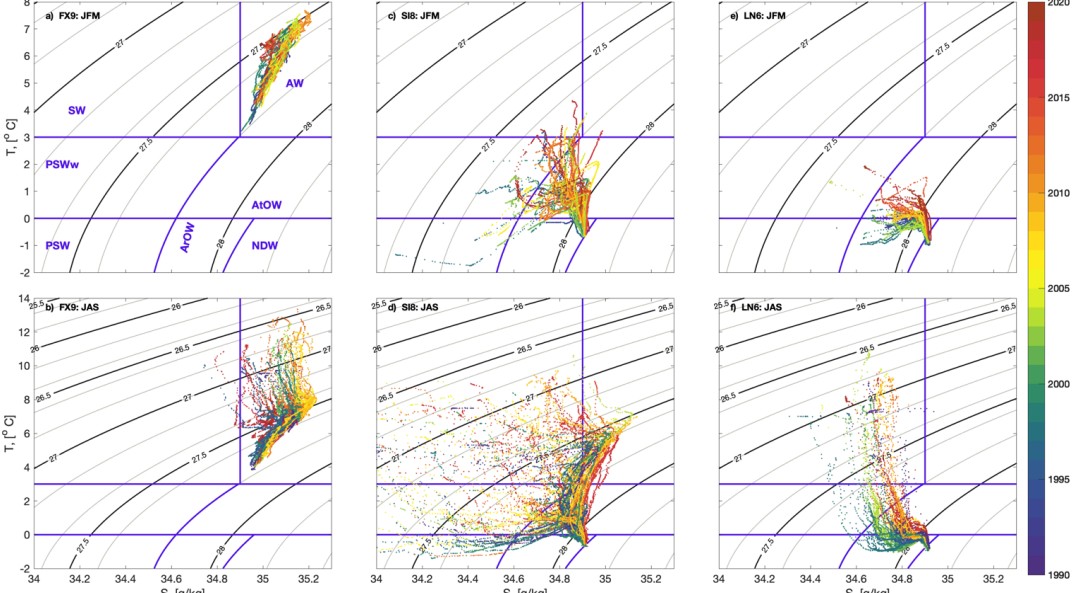

**Figure 2: (Top row) Winter (JFM) and (bottom row) summer (JAS) T-S diagrams for three selected stations (a, b) FX9, (c, d) SI8 and (e, f) LN6, considered as representative for the South, North and Northeast regions shown in Fig. 1. The T-S individual profiles are color-coded by year. The main water masses as defined in Table 2, are labeled in panel (a).**

The three stations have a clear seasonality. Overall, the summer profiles span a wider temperature range due to seasonal warming of the upper layers (Fig. 2b, d, f) than in winter (Fig. 2a, c, e). FX9 is notably warmer and saltier than the other stations, especially in summer (Fig. 2a), when the minimum temperature in FX9 (nearly 4ºC) is as high as the maximum temperature in SI8 and 2 degrees higher than in LN6 (Fig. 2a, b, c). At SI8 a large change in density between seasons is observed mainly driven by the contribution of AW, explained by offshore migration of the NIIC and the stronger inflow of

AW during the summer (Fig. 2c, and f). While the widest seasonal amplitude in salinity is observed at SI8, the largest seasonal amplitude in temperature is observed at LN6.

     FX9 does not show a clear interannual pattern in summer but in winter the 2000's strike out as saltier than the other years. In contrast, at SI8 and LN6 fresher and colder waters are observed in the 90's, they progressively warm and become saltier over time, and they reach their maximum temperature and salinity by 2015-2018. This decadal pattern is more evident in

winter, but it is observed in both seasons.

### 3.2 Seasonality of Stratification and Mixed Layer Depth (MLD)

     The spatial and temporal variability of the stratification around Iceland is remarkably large (Fig. 3 and Fig. 4), and the relative contribution of temperature and salinity shows a strong seasonal cycle. In summer, the MLD is relatively shallow, oscillating around 50 m with a small standard deviation (Fig. 3). In contrast, in winter the ML reaches depths greater than

400 m in FX9, ST5 and SB5 with large standard deviations spanning a 100 m range (Fig. 4). The deepest average MLD is found in FX9 while the shallowest are KG6, SI8, LA8 and KR6.

     In summer (Fig. 3), the upper-ocean stratification around Iceland (Fig. 3) is generally dominated by temperature, except for the three Northwestern stations (namely LB8, KG6, SI8). LB8 exhibits the largest variability in both $N_T^2$ and $N_S^2$, but is mainly dominated by salinity in the upper 200 m and by temperature below that depth. This transition station is located at





the sill of Denmark Strait, a convergence zone for several currents (see Table 1 and Fig. 1) carrying water masses with contrasting T-S properties within the ML and the thermocline (Jónsson, 1999; Logemann et al., 2013; Casanova-Masjoan et al., 2020). In KG6, the fresh inflow from the EGC compensates for the cold temperature, and salinity largely dominates stratification. For SI8, we observe a mixed regime with almost equal contributions from both salinity and temperature to the total stratification, which suggests that this is also an area of transitional regime. For the stations: LN6, LA8 and KR6,

despite stratification is mainly due to temperature, there is a small subsurface contribution of salinity just below the ML, likely due to the presence of fresh PSWw. The southern stations ST5 and SB5, have a minimal contribution from salinity, which may be associated with the numerous river discharges and the proximity to the continental shelf.

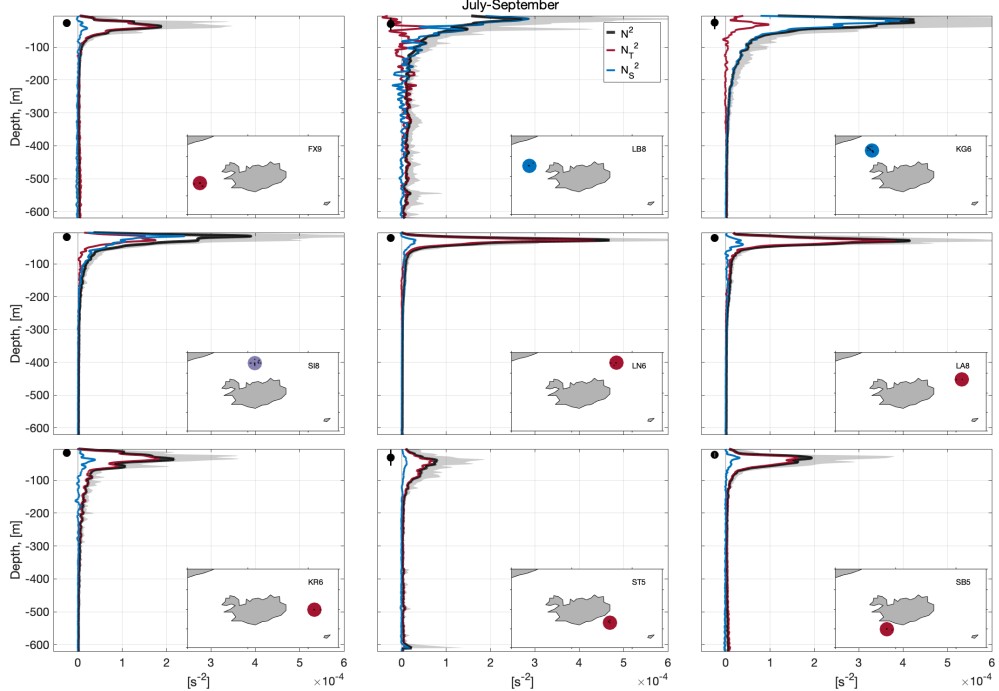

**Figure 3: Summer (JAS) average density stratification ($N^2$) profiles for the selected stations; the average total stratification (black)**
**is decomposed into temperature (red) and salinity (blue) contributions, while the gray shaded band represents all the stratification profiles. The black solid dot represents the average MLD with the error bar showing the standard deviation as an indicator of the temporal variability. The maps in the lower corner show the location of the station color coded by the dominating regime according to the contribution to stratification: red for temperature, blue for salinity, and purple for a mixed regime.**

The hydrographic onset is very different for winter; the stratification is one order of magnitude lower, i.e., comparing
Figure 3 and Figure 4. Also, the water temperature is much colder due to winter heat loss. Under these conditions, the relative contribution of salinity to the total stratification stands out around Iceland except at the southern stations (FX9, SB5, ST5), where the weakest winter stratification is observed. This southern region shows the deepest MLD, between 350 and 700 m in the stations FX9, SB5 and ST5, while for the northern stations (KG6, SI8, LN6, LA8, and KR6) the mean winter MLD is about 100 m. Similar to summer, station LB8, also shows high variability in winter stratification, associated
with the confluence of currents at the Denmark Strait.

The role of temperature or salinity in setting the stratification (α- and β- ocean, see *e.g.,* Carmack, 2007) is linked to the hydrographic characteristics (temperature and salinity) of the dominant water masses within each region. Based on this, we can classify the waters around Iceland. The southern side is an α-ocean as it receives the influence of warm and relatively



salty AW. Hence, the stratification is mainly temperature driven in both seasons (see FX9, ST5 and SB5 in Figures 3 and
4) and MLD gets deeper than 400 m in winter. The northwest of Iceland (LB8, KG6) is under the influence of the EGC
throughout the year bringing fresh PSW and PSWw into the area. Therefore, this area with winter MLDs of 100-150 m and
can be considered a β-ocean, with heat fluxes equivalent to the southern region but with stronger and salinity dominated
stratification blocking the potential of deep convection, i.e., this region does not have a mechanism to lose surface buoyancy
seasonally in the salinity component. In contrast, the northeastern Icelandic area (SI8, LN6, LA8 and KR6) shifts from β
in winter to a mixed α/β in summer. This is likely due to an offshore migration of the NIIC increasing the inflow of AW
(Jónsson and Valdimarsson, 2012; Casanova-Masjoan et al., 2020, their Figure 11). For instance, in winter, SI8 has a PSW
signature at the thermocline with salinity driving the stratification and a MLD of about 90 m (β-ocean), and in summer, the
NIIC brings warm AW to the upper layers of SI8 making the stratification similarly driven by temperature and salinity.
Overall, the north of Iceland exhibits the strongest summer stratification of the study area which results in very shallow
MLDs.

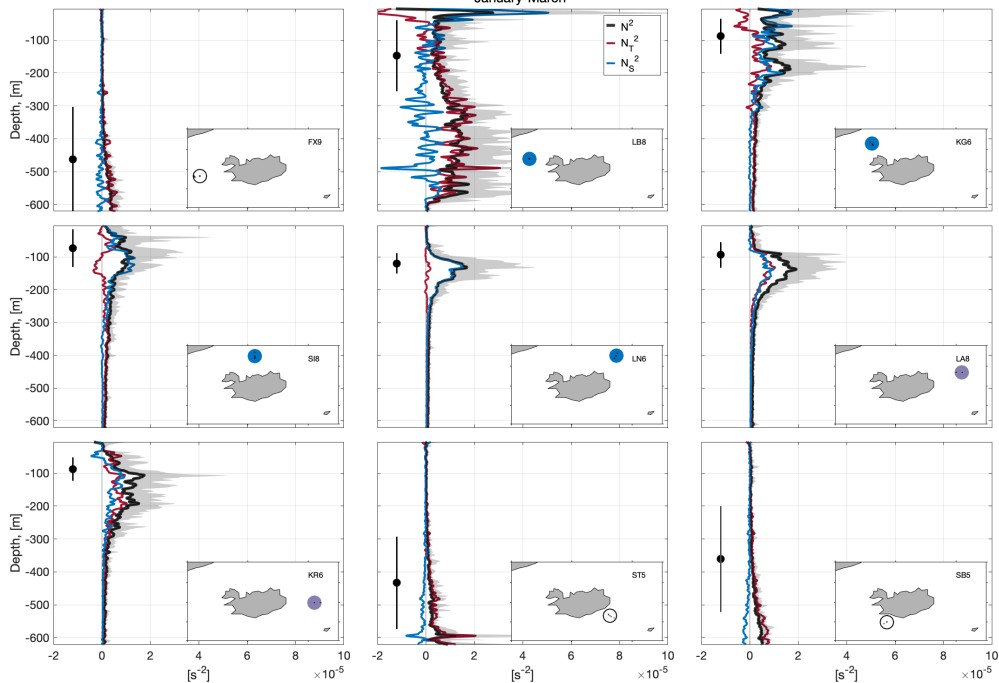

**Figure 4: Same as Figure 3 but for winter (JFM). The maps in the lower corner with a white circle indicates very weak winter stratification.**

**3.3 Interannual to decadal variability of the mixed layer properties**

To investigate the interannual to decadal ML variability we focused on three reference stations, considered representative
of the α-ocean (FX9, West), transition (SI8, North) and β-ocean (LB8, Northwest) regimes around Iceland. FX9 for an
α−ocean dominated by relatively warm and salty AW, SI8 as a transition area, and LB8 for a β-ocean dominated by cold
and fresh PSW. The three stations show strong interannual variability.

In FX9, to the West of Iceland, there is a high correlation between mixed layer temperature (MLT) and salinity (MLS)



anomalies. Between 1990 and 1998 the mixed layer was the deepest, coldest and second fresher period, as shown in Figure 5a, and d (positive MLD anomalies correspond to deeper ML and negative correspond to shallower ML). Around the period 2000-2014, there is an increase in MLT and MLS as the ML shows a moderate shallowing. The winter MLD is the shallowest, saltiest, and warmest in 2010 (Fig. 5a, and d), where the temperature contribution seems to control this

minimum. From 2015 to 2018 the ML returns to the cold and fresh conditions of the 90's but the MLD is average. The observed variability of the ML and its properties in FX9 correlates with the NAO (Fig. 5g, h). Positive NAO at the beginning and the end of the time series, corresponds with deeper colder and fresher MLs, while negative NAO between 2000 and 2015 roughly corresponds with shallower, warmer, and saltier MLs. As shown in Figure 2, FX9 contains only AW (Fig. 2a, d) likely advected from the south to the area by the Gulf Stream and later the Irminger Current. Similar conditions have

been observed in the Irminger Sea over the same period, and they have been related to the NAO phase and its impact on the Subpolar gyre (Feucher et al., 2022). This suggests that the FX region is largely influenced by the Atlantic climate and therefore it is impacted by the NAO (Bersch, 2002).

At SI8, in the North of Iceland (Fig. 5b, e, h), the negative winter MLD anomalies are in the order of those at FX9 and also exhibits strong interannual variability without an identifiable pattern. Strong positive MLD anomalies are observed in

particular years (*e.g.,* 2000, 2007, and 2016) but they do not seem correlated neither with the MLT/MLS nor with the NAO variability. Interestingly, the MLT and MLS co-vary during the period 1990-2005, when the mixed layer is colder and fresher, but this correlation weakens from 2005 to 2018, when the positive MLT anomalies increase while the MLS anomalies, although positive, do not vary significantly.

In LB8 (Northwest), the winter MLD has the largest variability as it is in the vicinity of the front between NIIC and the

EGC, which shapes the Polar and Atlantic conditions. Despite this large variability, a co-variance between MLT and MLS anomalies seems to be correlated with the position of the front, fresher and colder MLs associated with EGC influence and warmer/saltier MLs with the presence of NIIC (Fig. 5c, f, l). Generally, shallower MLs are also fresher and colder, which agrees with a salinity-dominated stratification in the upper layer (Fig. 4b). Three particular years present relatively deep, cold, and salty MLs: 1996, 2006 and 2014. The observed interannual variability in the ML and its properties, while large,

does not seem to be correlated with the NAO, except the last decade where the high state of NAO is consistent with the positive MLT and MLS suggesting a larger presence of the NIIC at this station.




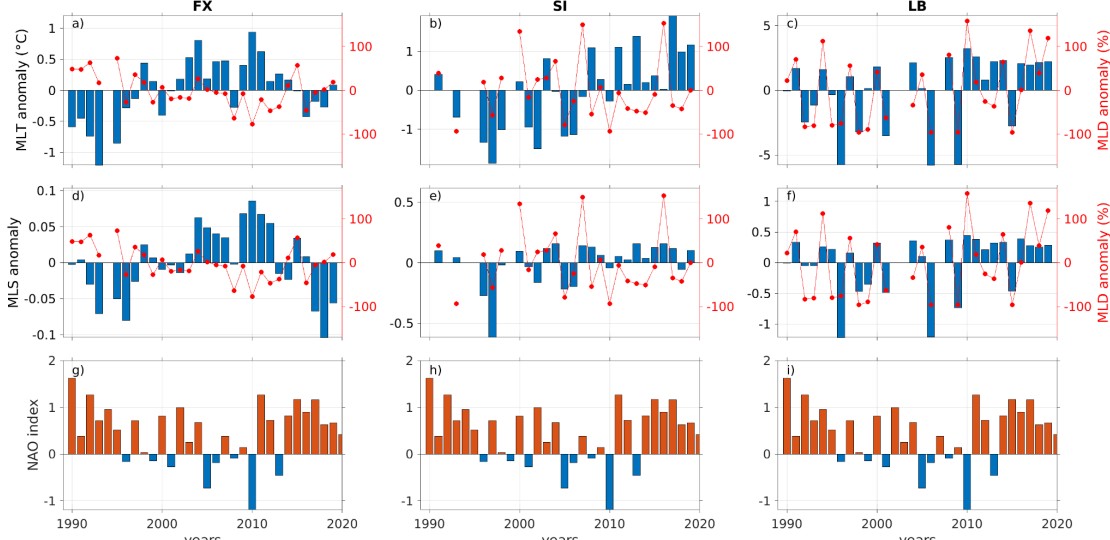

**Figure 5: Interannual winter (JFM) variability of the MLD anomaly with negative values associated with shallower MLs (red line), its temperature (MLT, first row) and its salinity (MLS, second row) from 1990 to 2018 in three stations representative of different regions around Iceland. (a, d) FX, in the Southwest, (b, e) SI, in the North, and (c, f) LB in the Northwest. The last row (g-i) shows the winter average of the NAO index for comparison. For MLD positive anomalies correspond to deeper ML, positive anomalies in MLT and MLS correspond to warmer and saltier water, respectively.**

To delve into the interannual to decadal variability of the MLT around Iceland, we analyzed its anomalies in all nine stations and computed their linear trends (Fig. 6). The temperature anomalies show significant interannual variability and spatial differences around Iceland. For instance, positive anomalies are observed in 2003 in most of the stations in both seasons, with particularly large temperature anomalies east of Iceland. Strong warm anomalies are also observed in 2017, mostly in summer at all stations except FX9 and SB5, located south of Iceland (Fig. 6; left panel). Although the 28-years period might be too short for identifying linear anthropogenically-driven trends, linear trends are significant in some of the stations, (where the p-value is indicated). The linear trends show a general warming of the mixed layer that is more evident in winter, mainly in the stations of the northeast (LN6, LA8). In the south (ST5, SB5 and FX9), even if a trend appears significant from 2000-2015, they seem to be aliased with interannual variability that induces colder mixed layer conditions from 2015 to 2018. This return to the conditions observed in the early 90's could be associated with the NAO (Future et al., 2022) as shown in Fig. 5. The observed general warming of the ML around Iceland is consistent with the progressive warming of the NIIC (Casanova-Masjoan et al. 2020, Polyakov et al. 2017).



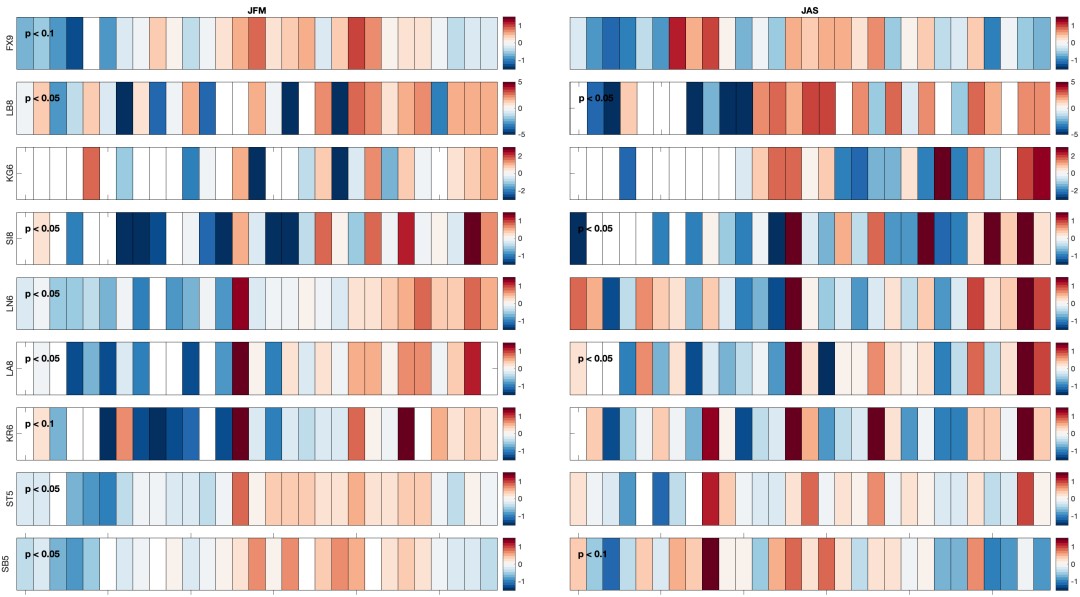

**Figure 6: Mixed Layer Temperature (MLT) anomaly time series (left) winter (JFM) and (right) summer (JAS) for the 9 stations shown in Figures 3, 4. The anomalies for all stations except LG8 and KG6 have fixed scale from -1.5 to 1.5 C.**

## 4. MLD driving mechanisms from a 1D model

The summer stratification around Iceland is an order of magnitude higher than in winter and hence summer MLDs are very shallow and with higher variability. Therefore, we only use the winter season to study the atmospheric effect on these α- and β-ocean regions by using the Price et al. (1986) one-dimensional model. It is worth mentioning that the 1D model estimates the MLD from temperature-based profiles while the estimates from observations are density threshold based. The MLDs shown in Figure 7 are within the range of the observed average ± standard deviations (black dots and lines in Figure

7). For all stations a spring shoaling of the MLD is driven by heat flux, while the MLD remains relatively deep due to the wind-stress.

In the model, the stations embedded in AW (SB5 and ST5) present the largest MLDs exceeding 300 m depth, which is consistent with the observations. In this α-ocean region, the development of a deep ML is driven mainly by heat fluxes (Fig. 7 h and i), which also holds for FX9. However, wind-stress steadily contributes to the development of the ML (Fig.

7a, h and i). During the summer, shoaling of the mixed layer is likely influenced by the changes of both heat and freshwater fluxes, with their effects on the MLD partially offset by wind stress (Fig. 7a, h and i).

The station LB8, despite being in Denmark Strait and presenting a large contribution of PSWw and PSW in the upper layers (driving a β-ocean stratification), shows that the development of MLDs can be influenced by both, heat flux and/or wind stress (Fig. 7b). However, the contribution of wind-stress leads to a slightly deeper MLD than the heat fluxes (Fig.

7b). Wind stress becomes the lead forcing mechanism northeast of LB8, coinciding with the shift from α- to β-ocean stratification (Fig. 7b, c and d). This region represented by LB8 station does not have a large convection potential compared to those with pure AW and hence do not produce large MLDs (Fig. 4) and convection is mainly driven by wind-stress.

At LN6, LA8 and KR6 before spring, wind-stress and heat fluxes contribute similarly to the development of the MLD, those stations are at the core of the EIC, which contains PSWw in the upper layers and AtOW beneath. Hence the wind-



stress and heat fluxes are the driving mechanisms acting on the water column (Fig. 7e and f). As the wind-stress develops, the MLD evolution erodes the PSWw strata reaching the AtOW layer, allowing the heat fluxes to lead the MLD development.

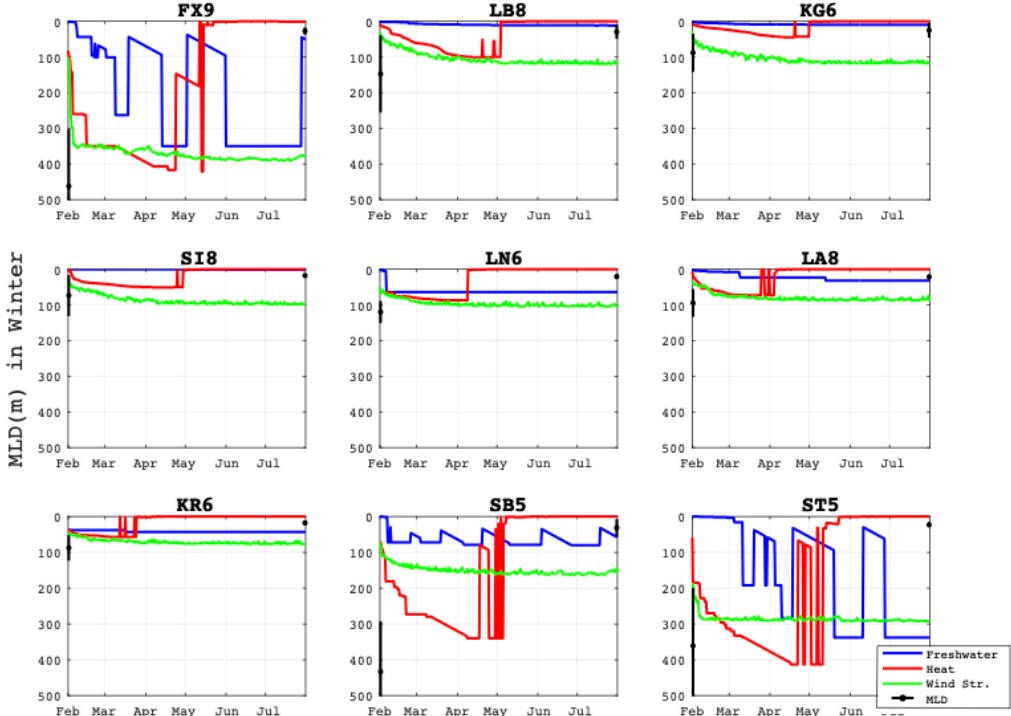

**Figure 7:** MLD driving mechanism decomposition estimated from the PWP 1-D model (Price et al. 1986) for each of the studied

stations. Different MLD evolutions are shown for outputs forced with freshwater fluxes (blue), heat fluxes (read), and wind-stress (green). Black dots represent the averaged winter and summer MLD with their standard deviations.

### 5. Stratification around Iceland

To complement the understanding of the stratification of the Arctic and Subarctic waters around Iceland, their connection with water masses, currents, and their variability, we used the spice frequency averaged in the first 200 m, estimated

following the methods described in Strehl et al. (2024). The distribution (Fig. 8) clearly reveals a southern (northern) side where the temperature (salinity) dominates the spiciness and hence marking clear $\alpha$- ($\beta$-) ocean regimes. These areas largely correspond with the distribution of AW versus PSW/PSWw (See Fig. 2 for T-S definitions).





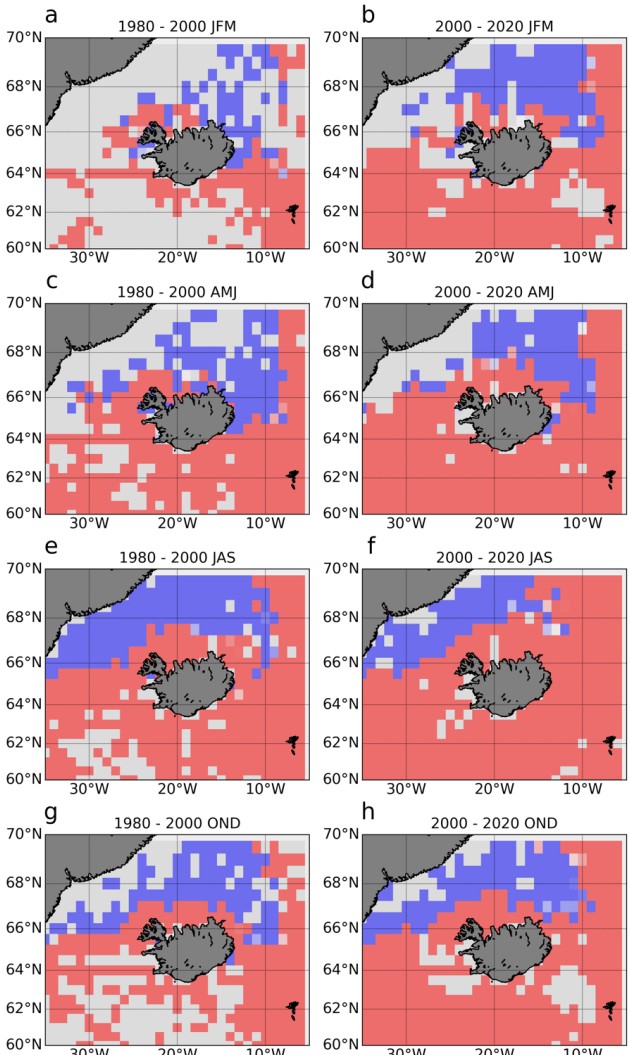

**Figure 8:** **Mean upper 200m spice frequency for the region of study showing the α-ocean (red) and β-ocean (blue) regions for the**
**periods of 1980-2000 and 2000-2020 and the four main seasons JFM (a,b), AMJ (c,d), JAS (e,f) and OND (g,h). The Brakstad et al**
**(2023) hydrographic dataset is used for this calculation.**

The seasonal distribution of spice frequency north of Iceland agrees with the seasonal behavior of the NIIC described in
Casanova-Masjoan et al. (2020) where the NIIC surface extension in winter and spring remains constrained to the northwestern
side of Iceland due to the cold southeast surface imprint of the EIC. Then in summer, the NIIC expands northeastward, reaching
all the way to the eastern side of Iceland and the northernmost station at the SI transect and constraining the polar water
between the northern end of the Kolbeinsey Ridge and the Greenland shelf. Then in fall the NIIC northern extension narrows
but is still able to surround Iceland. It is noteworthy that a clear increase in coverage of the α-ocean, mainly in summer, is
observed by comparing the 1980-200 average and the 2000-2010 average.

**6. DISCUSSION AND CONCLUSIONS**

In this study, we discussed the seasonal and interannual variability of the mixed layer characteristics and the stratification



regimes around Iceland by using a long timeseries of CTD data. Based on our results, we propose the regionalization of the waters around Iceland into three dynamical regions α-ocean, β-ocean, and transition-ocean.

The southwestern region is dominated by AW both in winter and summer. Within this region, the winter MLD is the deepest and most variable of the whole study area, with ML's occupied by AW over the whole sampling period. This region
is influenced by the dynamics of the Irminger Sea and the Subpolar gyre and has favorable conditions (α -ocean) for winter deep convection driven by heat fluxes to develop deep mixed layers, which agrees with previous studies (Carmack, 2007; Våge et al., 2008; Piron et al., 2016; Stewart and Haine, 2016; Petit et al., 2020). In the Southern region, the ML salinity anomaly was negative over the last 5 years, which is consistent with previous numerical models and Argo observations showing a freshening trend of the North Atlantic (Tesdal et al., 2018; Holliday et al., 2020, Liu et al., 2020). However,
since the South of Iceland is an α−ocean, these recent changes have not reached an effect the MLD.

The northwestern region, which includes the Denmark Strait, is a medley of all the water masses described in this study. In this region there is a confluence of Greenland shelf and slope waters and Iceland Sea origin waters (Harden et al., 2016; Foukal et al., 2020). This includes the NIIC generating an important variability in the water properties due to complex interactions of the regional currents (Lin et al. 2020, Mastropole et al. 2017). The MLD variability over time at the KG6
and LB8 stations is moderate (<100m), except for the years 2000, 2007 and 2016 when the ML was anomalously deep. In this region, the stratification is notably year-round dominated by salinity (β-ocean), which is explained by the strong Arctic influence of cold and fresh waters transported by the EGC. A broader look into the northwestern side of the basin reveals that this area can be divided at the center of the Denmark Strait into an α-ocean near the Icelandic shelf where the NIIC flows and a β-ocean as we progress towards Greenland (where the LB8 and KG6 stations are located). Near the Icelandic
shelf, MLDs are driven mainly by wind-stress, with a secondary contribution of heat fluxes. The T-S properties as well as the MLT anomalies in the northwest region near the Icelandic shelf show that the NIIC waters there are getting warmer and saltier. This agrees with previous studies showing the transformation of the NIIC also accompanied by an increase in its transport with time (Casanova-Masjoan et al., 2020). Even if the α-ocean area is warming, it is not expanding northwestward, hence the EGC is acting as a barrier bringing PSW in the area and maintaining the β-ocean state on the
northwesternmost side of the Strait.

Northeast of Iceland, intrusions of SW (and AW) have been detected in the winter (summer) ML. In this region, the stratification changes from β- to α-ocean seasonally. The Kolbeinsey Ridge acts as a barrier where we find the eastward penetration of the EIC bringing fresh waters (PSWw) from the East Greenland Current (Macrander et al., 2014; Casanova-Masjoan et al., 2020). In summer, the NIIC expands northeastward, bringing AW into the area and changing the
stratification regime to α. Hence, the mixed layer waters show important seasonal variability, they range from maximum temperature below 2 °C in winter to over 10 °C in summer. This is also the only region where a significant warming decadal trend emerges over the interannual variability and progressively results in a stronger α-ocean. This agrees with the AW warming observed in Casanova-Masjoan et al. (2020) and with the northward progression of AW named as 'Atlantification' described by Polyakov et al. (2017). This shift to α-ocean or 'Atlantification' may lead to deeper ML's and the associated
deeper convection may increase the potential of this area to contribute to the dense flow carried by the NIJ (Semper et al. 2019).

The regionalization proposed in this work, based on hydrographic properties, matches the recently proposed distribution of primary production around Iceland (Richardson and Bendtsen, 2021; Cerfonteyn et al., 2023), supporting the importance of MLD properties for the primary production (Ólafsson, 2003). The induced alterations on primary production can lead to
ecosystem changes. For example, Iceland has witnessed a rapid increase in the population of mackerel, a relatively warm-water fish, since 2006 (Astthorsson et al., 2012; Campana et al., 2020) starting from the Southeast towards the north and recently they have been reported almost all around the country. This migration is consistent with our observations of both,





the increase in surface temperatures, *i.e.,* northward shift of warmer isotherms over the Iceland Faroe Ridge (de Marez et al., 2025), and the increase of temperatures within the ML in the same regions over the last decade, which may establish

new pathways for entire ecosystems.

The long time series investigated here revealed important interannual oscillations of the ML properties. Four main features are to be highlighted: (i) We do not observe any linear trend in the MLD, which is rather subjected to strong interannual variability (ii) Except for the southern stations, influenced by the subpolar gyre, the interannual variability was not correlated with the NAO. (iii) The linear fit indicates significant (at 95%) warming trends in the MLT of most of the

stations in winter, at times reaching 2 °C. This agrees with previous studies (Sarafanov, 2009) showing that the northern part of the North Atlantic (south of Iceland) is strongly dominated by atmospheric interannual to decadal variability, particularly, where AW is present. The exception here is the Northeastern region of Iceland where we observe a clear warming trend of the ML (2010-2020). (iv) We observe an 'Atlantification' expressed as a northeastward progression of the α-ocean state. This progression will highlight the role of the northeastern area of Iceland as a convective zone where

deep water could be formed and contribute to the NIJ.

## AUTHOR CONTRIBUTIONS

Conceptualization, ARA, MDPH and EP; methodology, ARA, EP and MDPH; software, ARA and AP; formal analysis, ARA, EP, TM, CdM and MDPH; investigation, ARA, EP and MDPH, AM; data acquisition, SRÓ and AM; data curation, SRÓ and AM; writing—original draft preparation, ARA, EP and MDPH; writing—review and editing, SRÓ, AM and SJ,

TM; visualization, ARA and EP; project administration, SRÓ; funding acquisition, ARA and SRÓ. All authors have read and agreed to the published version of the manuscript.

## FUNDING

ARA has been supported by HM Queen Margrethe II´s and Vigdís Finnbogadóttir's Interdisciplinary Research Centre on Ocean, Climate and Society (ROCS) under grant no. 158-4223. Support for this work was also provided by the European

Union's Horizon 2020 research and innovation programme under grant no. 727852, Blue-Action project (AM and SJ). This work has been supported by the FAR-DWO (PID2020-114322RBI00) project from the Spanish Ministry of Research.

## ACKNOWLEDGMENTS

We are grateful for the invaluable cooperation we have had with the crews of the Icelandic research vessels Bjarni

Sæmundsson and Árni Friðriksson and to the many people at the Marine Research Institute that have contributed to the hydrographic observations over the years.





**Table 1.** Characteristics for the representative stations for each typical surveyed section. The representative ocean currents
at each section are also shown: North Icelandic Irminger Current (NIIC), Irminger Current (IC), East Greenland Current
(EGC), North Icelandic Jet (NIJ), East Icelandic Current (EIC), and North Atlantic Current. (NAC).

| Station | Depth, m | Lon | Lat | Oceanic region | Significant currents |
|---|---|---|---|---|---|
| Faxaflói (FX9) | 1010 | -27.98 | 64.35 | Subpolar North Atlantic | IC |
| Látrabjarg (LB8) | 658 | -27.050 | 66.083 | Denmark Strait | NIIC, EGC, DSO |
| Kögur (KG6) | 980 | -23.933 | 67.583 | Greenland Sea | EGC, DSO |
| Hornbanki (HB6) | 612 | -21.583 | 67.50 | Greenland Sea | NIIC, NIJ |
| Siglunes (SI8) | 1023 | -18.83 | 68.00 | Kolbeinsey Ridge | NIIC, EGC, EIC, NIJ |
| Langanes NE (LN6) | 1850 | -12.66 | 68.00 | Iceland Sea | EIC |
| Langanes E (LA8) | 1251 | -9.00 | 66.37 | Iceland Sea | EIC |
| Krossanes (KR6) | 1419 | -9.00 | 65.00 | Iceland Sea/ North Atlantic | EIC, NAC |
| Stokksnes (ST5) | 1153 | -13.66 | 63.66 | North Atlantic | NAC |
| Selvogsbanki (SB5) | 1006 | -21.48 | 62.98 | North Atlantic | IC |

**Table 2.** Main water masses definitions for the region of study (Rudels et al., 2005; Våge et al., 2011)

| Water mass | Potential Temperature ($\theta$) | Salinity | Potential density ($\sigma_0$, kg m$^{-3}$) |
|---|---|---|---|
| Surface Water(SW) | $< 3^\circ C$ | - | $\sigma_0 < 27.70$ |
| Warm Polar Surface Water (PSW$_w$) | $0^\circ C < \theta < 3^\circ C$ | - | $\sigma_0 \geq 27$ |
| Polar Surface Water (PSW) | $< 0^\circ C,$ | - | $\sigma_0 < 27.70$ |



| | | | |
|---|---|---|---|
| Atlantic Water(AW) | $> 3°C$ | $> 34.9$ | - |
| Atlantic-origin Overflow Water (AtOW) | $0°C < \theta < 3°C$ | - | $\sigma_0 < 27.8,$ <br> $\sigma_{0.5} < 30.44$ |
| Polar intermediate Water (PIW) | $0°C$ | $\leq 34.676$ | $\sigma_0 > 27.70,$ |
| Arctic-origin Overflow Water (ArOW) | $< 0°C$ | - | $\sigma_0 > 27.8,$ <br> $\sigma_{0.5} < 30.44$ |
| Nordic Seas Deep Water (NDW) | $< 0°C$ | - | $\sigma_{0.5} \geq 30.44$ |









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
