# Peer review of "Stratification and Mixed Layer Depth around Iceland, characterization and inter-annual variability"

_EGUsphere, 2025_

## Referee Comment (RC2)

**Stratification and mixed layer depth around Iceland, characterization and inter-annual variability**

by A. Ruiz-Angulo, E. Portela, C. de Marez, A. Macrander, S. R. Ólafsdóttir, T. Meunier, S. Jónsson, and M. D. Pérez-Hernández

In this manuscript, the rich hydrographic data set collected through long-term monitoring efforts is used to characterize stratification and mixed-layer properties around Iceland. Three distinct regions are identified, characterized as α-, β-, or transition-oceans based on whether temperature or salinity dominates stratification, and implications for water mass transformation are discussed. Straddling the Greenland-Scotland Ridge, Iceland is situated at a key location for the overturning circulation in the Atlantic Ocean, particularly as regards the supply of dense water masses to its lower limb. I think this is an interesting and timely contribution to improve our understanding of the unique importance of Icelandic waters, but I have some concerns about the determination of mixed-layer depths, water mass definitions, and the one-dimensional mixed-layer model simulations that I hope the authors will address.

**Major comments:**

Your method for determining mixed-layer depths, with an adjusted density criterion, is commonly used. Verifying the outcome of the routine by visual inspection is a good way to ensure that the results are realistic. However, I think that you also need to document this, or at least be more specific about what the mixed layer depth you arrive at represents. Is it the surface mixed layer that is currently in direct contact with the atmosphere or the deepest weakly-stratified layer formed the same winter that may better represent the depth that convection has reached? In the subpolar North Atlantic and the Nordic Seas stacked mixed layers are prevalent and automatic routines are generally not very adept at identifying the depth of wintertime convection (Pickart *et al.*, 2002; Våge *et al.*, 2015; Brakstad *et al.*, 2019). I encourage the authors to consider additional verification of the mixed-layer depths and expand the discussion about the limitations of the routine and what the outcome actually represents.

More details about the PWP model (Price *et al.*, 1986) should be included in sections 2 (Data and methods) and 4 (MLD driving mechanisms from a 1D model). In particular, I miss information about how the model functions, about the initial conditions and forcing time series, and about the integration period and time step used for the simulations. I am sceptical about some of the results from the simulations, in particular the substantial impact of freshwater forcing on stations FX9 and ST5. Freshwater fluxes are usually of lesser importance for the mixed-layer development and tend to restratify the mixed layer (i.e., have a stabilizing influence) (e.g., Brakstad *et al.*, 2019), yet on these two stations they appear to cause a mixed-layer deepening of more than 300 m, which would require substantial evaporation in a region where precipitation generally dominates. At these two stations, wind-mixing is also surprisingly deep. Please confirm that the model simulates the mixed-layer development correctly and expand the discussion about these surprising results.

I am not fully convinced that these water mass definitions (Figure 2, Table 2), dating back to Rudels *et al.* (2005) and primarily focused on polar conditions, have high fidelity around Iceland today. For example, the 0°C isotherm no longer provides accurate distinction between Atlantic- and Arctic-origin water masses (Våge *et al.*, 2022). Perhaps these definitions should be updated or adapted for current conditions around Iceland? It matters when these definitions, without additional evidence, are used to determine the origins of different water masses. For example, are you confident that the dominant water masses near the surface at SI8 are of polar origin (line 172)? At times, the cold, fresh profiles in Figures 2c and 2d are unmistakably polar (compare

to Figures 11c and 12c in Rudels *et al.*, 2005), but while there may be some trace of polar water at SI8 the rest of the time I am not convinced that it is the predominant surface water mass. The same applies to the Atlantic-origin water at the same station (line 173). This could be more recently-formed intermediate water in the Iceland Sea that may be warmer than 0°C (Våge *et al.*, 2022). A closer inspection of each profile, to verify that it displays the intermediate temperature- and salinity maxima characteristic of Atlantic-origin water (Mauritzen, 1996), would likely be necessary for unambiguous identification.

**General comments:**

The paragraph on Atlantification of the Arctic Ocean in the introduction (line 63) appears unmotivated and its purpose is not immediately clear. The transition into an α-ocean north of Iceland is later ascribed to Atlantification, though Atlantification is more generally considered where Atlantic properties displace Polar properties in the Arctic Ocean. If applicable north of Iceland, I think you need to start making a stronger case for the relevance of Atlantification already in the introduction and follow that up in the discussion (e.g., line 373).

Why did you limit the data set to include observations only to 2019 (line 105)? The MFRI monitoring program is ongoing, so more recent data should be available. You document interesting signals toward the end of the record (Figure 5) that could be further examined.

Did you consider including other stations than the deepest on each transect in the analysis? At least some of the transects have several stations offshore of the shelf break. Perhaps for one transect in each hydrographic region, it would be very interesting to know if the other deep stations corroborate the results – that would make your conclusions more robust.

While LB8 may be representative for a β-ocean regime, located in the middle of Denmark Strait it is also characterized by substantial high-frequency variability (von Appen *et al.*, 2014; Lin *et al.*, 2020). Perhaps KG6 would be a better choice for investigating interannual to decadal variability in mixed-layer properties in this region?

I think the text would be more clear if the past tense was consistently used for past events (e.g., line 248). At least, avoid switching between past and present tenses within the same paragraph.

The NAO primarily represents atmospheric variability on time scales shorter than the ocean may fully adjust. Variability in circulation and mixed-layer depth/properties may be expected (e.g., Yashayaev, 2007), but to a lesser extent wholesale changes throughout the water column, hence line 257 could be phrased more conservatively.

If the mean mixed-layer depth at SI6 is less than 100 m (Figure 4), how can anomalies in mixed-layer depth exceed 100 m in both positive and negative directions (Figure 5b, e)?

For the statistical analyses (Figures 5 and 6), please provide the values for the correlations and trends reported in the text as well as their uncertainties. That would help make the analysis more quantitative and robust.

Please provide some information about the gridding or averaging used for the data set that you calculate the spice frequency shown in Figure 8 from.

**Detailed comments:**

Line 16:
Dense-water formation is generally considered a necessary condition for a global overturning circulation rather than a driving mechanism (e.g., Kuhlbrodt *et al.*, 2007).

Line 24:
Contributions should be in plural.

Line 40:
As a proper name, ridge must be capitalized in the Greenland-Iceland-Scotland Ridge.

Line 44:
The Faroe Islands would be more appropriate than just Faroe.

Line 52:
Regarding the transformation of AW into AtOW, it is unclear what is meant by "along the Norwegian Current western intrusions". Presumably that water mass transformation occurs along the Norwegian Current, which is a main mode of water mass transformation that is otherwise not mentioned here (e.g., Mauritzen, 1996; Huang *et al.*, 2023).

Line 56:
Very likely heat loss is important also for water mass transformation east of Greenland, but wind forcing and sea-ice retreat are necessary for preconditioning this region before convection can occur.

Lines 60:
Do you mean from the Arctic Ocean? The Arctic, by itself, is ill-defined and has multiple interpretations. The reference to Arctic as north and northwest of Iceland (line 91) does not help clarifying.

Line 64:
The word "extent" is used twice in the same sentence.

Line 71:
Please be more specific, under what conditions does wind forcing enhance turbulent mixing that deepens the mixed layer? The following sentence does not add clarification regarding the impact of wind forcing, even though it alludes to an example of this process.

Line 79:
The definite article in front of 40% should be removed.

Line 82:
It would be good to be more specific about what kind of mixing you refer to, perhaps diapycnal mixing or convection?

Line 170:
There should be a comma after southwest of Iceland.

Line 185:
A reference to Jónsson and Valdimarsson (2012) would be appropriate here.

Line 214:
What do you mean by "hydrographic onset"?

Line 226:
The sentence starting with "Therefore, ..." is unclear and should be rephrased.

Line 228:
It should be: "... the potential **for** deep convection ..."

Line 241:
The paragraph starting on this line is repetitive and could be written more concisely.

Line 256:
The Feucher et al. (2022) paper is missing from the reference list.

Line 278:
Please specify what the anomalies are computed from. Are they anomalies relative to the record-long mean at each station?

Line 286:
The term aliasing has a specific meaning in time series analysis, where high-frequency variability appears as a lower-frequency signal due to inadequate sampling. It is not clear that the term was used appropriately in this sentence.

Line 289:
The reference to Polyakov *et al.* (2017) is not appropriate here, they did not consider the mixed layers around Iceland or the NIIC in their study.

Line 295:
Mixed layers in summer are shallow primarily because the buoyancy forcing abates in spring and becomes largely positive in summer, episodic wind and wave mixing (also reduced in summer) ensure that there is a mixed layer. This is a main reason why stratification is higher in summer than in winter. Please rephrase.

Line 311:
Do you mean SI8 rather than LB8, if you refer to the region north of Iceland?

Line 312:
Wind-stress, by itself, will not cause convection - for that you need buoyancy forcing. This process would more appropriately be referred to as wind-mixing.

Line 350:
There is at least a preposition missing near the end of this sentence.

Line 356:
The surface water masses of the EGC would have a strong Polar (not Arctic) influence.

Line 375:
An increasingly temperature-dominated stratification may in principle lead to deeper convection, but reduced atmospheric forcing and shallower, less dense mixed layers (Moore *et al.*, 2015; Våge *et al.*, 2022) strongly suggest that the Iceland Sea is not likely to become an important source to the NIJ.

Line 390:
The warming trend of 2°C, was that over the entire record length? When reporting a trend, please also provide a time reference.

Figure 5:

It would have been illuminating to see panels of mixed-layer density anomalies as well.

Figure 6:

I think it would be advantageous to use the same color scale for all panels, even if that means that some colors occasionally will saturate, to facilitate direct comparison between stations.

Figure 6, caption:

It should be: "... all stations except **LB8** and KG6..."

Figure 7:

Please also show the total mixed-layer depth resulting from the combination of all driving mechanisms. Do they add linearly? While the observations stem from February and not end of winter when the mixed-layers are deepest, it appears that some of the simulated MLDs will be substantially deeper. In the text, individual panels are referred to by letters, but these letters are not visible on the figure.

Table 1:

KG6 and HB6 are not located in the Greenland Sea. Referring to this region as Blosseville Basin or western Iceland Sea would be more appropriate.

Table 2:

Some of the less than/greater than signs are in the wrong direction and contradict Figure 2 (e.g., SW temperature, warm PSW density, and AtOW density).

**References**

Brakstad A, Våge K, Håvik L, Moore GWK. 2019. Water mass transformation in the Greenland Sea during the period 1986-2016. *Journal of Physical Oceanography* **49**: 121–140, doi:10.1175/JPO–D–17–0273.1.

Huang J, Pickart RS, Chen Z, Huang RX. 2023. Role of air-sea heat flux on the transformation of Atlantic Water encircling the Nordic Seas. *Nature Communications* **14**: doi:10.1038/s41 467–023–35 889–3.

Jónsson S, Valdimarsson H. 2012. Water mass transport variability to the north Icelandic shelf, 1994-2010. *ICES Journal of Marine Science* : doi:10.1093/icesjms/fss024.

Kuhlbrodt T, Griesel A, Montoya M, Levermann A, Hofmann M, Rahmstorf S. 2007. On the driving processes of the Atlantic Meridional Overturning Circulation. *Reviews of Geophysics* **45**: RG2001, doi:10.1029/2004RG000 166.

Lin P, Pickart RS, Jochumsen K, Moore GWK, Valdimarsson H, Fristedt T, Pratt LJ. 2020. "kinematic structure and dynamics of the denmark strait overflow from ship-based observations". *Journal of Physical Oceanography* **50**: 3235–3251, doi:10.1175/JPO–D–20–0095.1.

Mauritzen C. 1996. Production of dense overflow waters feeding the North Atlantic across the Greenland-Scotland Ridge. Part 1: Evidence for a revised circulation scheme. *Deep Sea Research I* **43**: 769–806, doi:10.1016/0967–0637(96)00 037–4.

Moore GWK, Våge K, Pickart RS, Renfrew IA. 2015. Decreasing intensity of open-ocean convection in the Greenland and Iceland Seas. *Nature Climate Change* **5**: doi:10.1038/nclimate2688.

Pickart RS, Torres DJ, Clarke RA. 2002. Hydrography of the Labrador Sea during active convection. *Journal of Physical Oceanography* **32**: 428–457.

Polyakov IV, Pnyushkov AY, Alkire MB, Ashik IM, Baumann TM, Carmack EC, Goszczko I, Guthrie J, Ivanov VV, Kanzow T, Krishfield R, Kwok R, Sundfjord A, Morison J, Rember R, Yulin A. 2017. Greater role for Atlantic inflows on sea-ice loss in the Eurasian Basin of the Arctic Ocean. *Science* **356**: 285–291, doi:10.1126/science.aai8204.

Price JF, Weller RA, Pinkel R. 1986. Diurnal cycling: Observations and models of the upper ocean response to diurnal heating, cooling, and wind mixing. *Journal of Geophysical Research* **91**: 8411–8427, doi:10.1029/JC091iC07p08 411.

Rudels B, Björk G, Nilsson J, Winsor P, Lake I, Nohr C. 2005. The interaction between waters from the Arctic Ocean and the Nordic Seas north of Fram Strait and along the East Greenland Current: Results from the Arctic Ocean-02 Oden expedition. *Journal of Marine Systems* **55**: 1–30.

Våge K, Moore GWK, Valdimarsson H, Jónsson S. 2015. Water mass transformation in the Iceland Sea. *Deep Sea Research I* **101**: 98–109, doi:10.1016/j.dsr.2015.04.001.

Våge K, Semper S, Valdimarsson H, Jónsson S, Pickart RS, Moore G. 2022. Water mass transformation in the Iceland Sea: Contrasting two winters separated by four decades. *Deep Sea Research I* : doi:10.1016/j.dsr.2022.103 824.

von Appen WJ, Pickart RS, Brink KH, Haine TWN. 2014. Water column structure and statistics of Denmark Strait Overflow Water cyclones. *Deep Sea Research I* **84**: 110–126, doi:10.1016/j.dsr.2013.10.007.

Yashayaev I. 2007. Hydrographic changes in the Labrador Sea, 1960-2005. *Progress in Oceanography* **73**: 242–276, doi:10.1016/j.pocean.2007.04.015.

---

## Author Comment (AC1)

**Reviewer 1**

This manuscript takes advantage of a long-term hydrographic data set from a series of stations encircling Iceland to assess major characteristics of upper-ocean variability and to evaluate long-term trends in the region. The study shows that salinity governs stratification at stations northwest of Iceland, while temperature governs stratification to the south. To the north of Iceland, alternating impacts from the North Icelandic Irminger Current and the East Icelandic Current lead to a mixed response.

The data set used for the study offers a rich supply of information, and the authors have chosen an interesting question to pursue. Parts of the analysis would benefit from more detail. I feel that the manuscript will likely be suitable for publication after careful revision.

1. The overall analysis of the manuscript addresses several distinct issues that are not always tightly linked together. One focal point is temperature vs salinity controls on mixed-layer depth or stratification, including consideration of the seasonal cycle. A second thread considers long-term trends in mixed-layer temperature and salinity along with the quantities with which they correlate. A third aspect assesses the multi-year linear trends in mixed-layer temperature in summer and winter. Analyses explore the mixed-layer evolution on the seasonal scale using a one-dimensional mixed-layer model and look at historic temperature vs salinity domination on a regional and seasonal scale. These are interesting and related analyses, but they are not fully linked together to provide clear and targeted interpretation of the results. For example, the long-term temperature trends in Figure 6 are interesting but not well connected with the rest of the analysis. In a rewriting, the manuscript should be more tightly focused to identify clear and linked results, well grounded in robust statistics.

We thank the reviewer for the valuable suggestions, which have helped us improve the quality of the manuscript. Our new version of the manuscript tries to link together those three stories, including the results presented in Figure 6.

2. One gap in the manuscript is a lack of statistical detail. This gap is particularly noticeable in Figure 5, in which the authors show time series of mixed-layer temperature and salinity anomalies, mixed-layer depth, and the North Atlantic Oscillation. The authors discuss correlations between these records but do not report correlation coefficients or statistical significance. To show that the patterns that the authors observe in their plots are robust, they should report quantifiable statistical metrics.

We agree with the reviewer. We have now computed Pearson correlations between all variables, and the results are detailed in Table 1.

Table 1: Pearson correlations between different variables. Non significant correlations have been omitted. The shown correlations are significant at 95% confidence (p

4. Given the discussion in Figure 5 and given the character of the records in Figure 5, I was surprised by the decision to fit trends in Figure 6. The discussion in Figure 5 emphasizes the specific relations between plotted quantities rather than long-term trends, so I was expecting Figure 6 to report correlations. It would be interesting to see the correlations between NAO, mixed-layer depth, and MLT mapped out for the full set of stations.

We thank the reviewer for this suggestion. In this study, one of the objectives was to address the long term variability of the seawater properties. In this sense, our choice was to address both, linear trends, as well as interannual variability. The two figures are then complementary., For figure 5, we chose representative stations, to analyse the interannual variability and its possible link with climate modes, but the correlation with the NAO is rather weak in most of them, it is only significant in the westernmost stations. Figure 6 would only expand on a question that we find to be already answered with Figure 5

Figure 6 is then focused on linear trends. Despite the presence of interannual variability on some of the stations, and even if the length of the record is not long enough to clearly detect anthropogenic trends, we do observe significant linear trends (particularly in winter) appearing over the interannual variability. This suggests that these should be explored in more detail in future studies, when longer time series will be available. This the message of Figure 6.

5. Since the analysis of Figure 6 focuses on trends, and the overall goals of the manuscript are directed toward alpha and beta oceanic regimes, the authors could/should expand the manuscript discussion to indicate how the trends (and regression coefficients, perhaps) inform their understanding of alpha vs beta ocean regions.

We thank the reviewer with this follow up over Figure 6. By adding the trends to each station, we can now argue that a transition into an alpha-ocean within the ML is underway. This signal exhibits higher statistical significance over the winter. The new version of Figure 6 clearly shows the statistically significant trends, supporting the interpretation of an ongoing transition toward an alpha-ocean regime around Iceland, suggested by Figure 8.

6. Line 133. "great score". In this usage, "great" sounds like a word that expresses an opinion. This point needs to be quantified, and more neutral wording should be used to express the skill of the density threshold method.

We agree with the reviewer, we have now rephrased the sentence using neutral wording, it now reads "...it shows to be effective even for cases..."

7. Lines 206-207. "The southern stations ST5 and SB5, have a minimal contribution from salinity, which may be associated with the numerous river discharges and the proximity to the continental shelf." This is an interesting point. Does the fresh water budget support this hypothesis? It would be useful to quantify the volume of freshwater discharge and its expected impact on salinity. Precipitation or oceanic circulation would be other factors that could influence salinity.

We have revisited this sentence and we believed that we did not look carefully at the data and that statement does not fully support this hypothesis. ST5 and SB5 are fully immersed in Atlantic waters, their TS diagrams look very similar to Figure 2a,b. The freshwater discharge around Iceland has characteristic peaks in January, May, and September with the maximum values in the South West (Reference Figure 2,4 Whitney, 2025).

Figure AR2. (Left) Annual mean near-surface salinity and surface currents around Iceland; (Right) River discharge time series for each of the 4 quadrants (Whitney, 2025).

The contribution of this fresh water seems to generate a small summer halocline, which is observed only in Figure 3 SB5 station. The winter convection is capable of eroding this small contribution to stratification. The winter and summer profiles for temperature and salinity for ST5 and SB5 are shown below and it is possible to observe the top small freshening (upper ~20m), which then disappears in the winter (Figure AR3). Moreover, SB5 is really close to the largest river discharge in Iceland, the South West. We have now rephrased the sentence adding the reference to the river discharges and explicitly mentioning the contribution to stratification.

Figure AR3. (top) Winter/summer vertical profiles for temperature and salinity for SB5 and (bottom) ST5.

8. Figure 3. The station labels are much smaller than the other figure labels and are too small to read clearly. The figure should be redrafted with larger labels.

We agree with the reviewer, the labels are too small. We have now fix this in both Figure 3 and 4.

9. Line 214. "hydrographic onset". The meaning of this is unclear. Does it refer to the top of the hydrographic profile or the seasonal onset of a change in the hydrographic profile?

Thanks for pointing this out, we mean that the state of the ocean is a lot different in the winter compared to the summer. We have now rewritten the sentence and it now reads: "The hydrographic conditions are very different for winter ...."

10. Figure 4. The figure shows open circles for deep mixed layers. The justification for this is not clear, since deep mixed layers can be as dynamically relevant as shallow mixed layers. Further explanation is needed.

We apologize for the confusion, perhaps both our figure and the caption were misleading. Figure 4 shows the MLD and the stratification decomposition of the average profiles for 9 stations. We do not have open circles for the deep mixed layers, the MLD are represented

by the solid black circles at the far left of each box. The open circles in Figure 4 represent the region where the data shows no significant dominance on the stratification decomposition, i.e., neither alpha nor beta oceans. In the revised version of the manuscript, we have corrected the caption of Figure 4 to clarify this point, explicitly explaining that the colorless or open circles correspond to areas with extremely weak stratification, where neither temperature nor salinity dominates. This clarification should help to avoid further confusion.

11. Line 270. Correlations with the NAO should be quantified. As noted above, the manuscript should report correlation coefficients and evaluate statistical significance.

The correlations have been quantified and they are now reported in the manuscript.

12. Figure 5. The repetition of panels g, h, and i seems unnecessary. Could the NAO time series be superimposed on the panels above (along with the addition of concrete correlation statistics)?

We completely agree with the reviewer. We have modified the figure following your recommendations:

13. Line 286. "aliasing". The term "aliasing has a specific meaning in time series analysis, and the usage here seems inconsistent with that usage. This could be described as "superimposed on".

We agree with the reviewer, aliasing is not a suitable word here and it has been modified in the revised manuscript.

14. Lines 295 and following. Choice of one-dimensional model. The Price-Weller-Pinkel model has been used extensively over the last four decades for upper ocean analyses. It is not the only possible model, and other recent studies have made use of GOTM or a stripped-down form of KPP. Thus, it's important to justify the choice of the PWP model.

We used the PWP model because, as you said, it not only has been extensively used in the Arctic and Subarctic regions but also because it is a simple 1D model that does not take into consideration advection or advected mix-layers. Hence, it is ideal to understand how local processes such as heat/freshwater fluxes or wind modify the vertical profile. Similar results were obtained using a ROMS model in the area (paper in preparation), therefore we did not explore other methods.

15. Lines 295 and following. The focus of the one-dimensional mixed layer analysis on winter only also needs clarification and should be more carefully described to explain that the analysis is really looking a the winter-to-summer transition. The PWP model has previously been used over a broad range of latitudes and for all seasons. Thus, a priori, there's not an obvious reason to exclude summer.

We excluded summer mainly because the MLDs developed are quite shallow, and even if it is forced with the atmospheric summer-to-fall- inputs shown on the supplementary material, the average profile does not end up producing the winter MLD in the PWP model. See figure below:

We have added a sentence in the revised version of the manuscript mentioning that the summer stratification is too shallow to appreciate the decomposition of processes.

16. Figure 7 calculations. How is the mixed-layer model initialized? Does it start with stratification typical of February? It's surprising that the mixed layer in the model appears to deepen at the outset. I would have expected it to be initialized with a profile that matches the climatological observations. This should be explained.

The 1D ML model is initialized with: freshwater flux, heat flux (each component and the sum) and wind speed obtained from ERA 5 (Copernicus). These parameters have been interpolated to each of the standard MFRI stations. In addition we used the CTD averaged summer and winter profiles at each station. All of these inputs are now shown in Supplementary material and Section 2 has been modified accordingly as follows:

Section2: "To investigate furthermore the driving mechanism of the MLD we used a one-dimensional model (Price et al., 1986) initialized with ERA-5 12-hourly dataset of wind stress, heat, and freshwater fluxes (Hersbach et al., 2020) and the summer/winter averaged vertical profiles of temperature and salinity from the observations presented here (see supplementary material). The 1D model would reveal the contribution from diurnal heating/cooling freshwater fluxes and wind mixing".

Figures added in supplementary material:

17. Figure 7 color scale. Please check colors. Green/red contrasts can be challenging for readers with limited color vision.

To avoid problems with colors we have decided to add different line styles to the new plot.

18. Line 389. "not correlated with the NAO". The lack of correlation should be quantified in the main body of the text, particularly if it is referenced in the conclusions. Better language would specify that the correlation with the NAO is not statistically different from zero.

Following both reviewer's suggestions, correlations have been quantified and this information has been added to the manuscript, which now reads: "Except for the southern stations, influenced by the subpolar gyre, the interannual variability was not correlated with the NAO. For example, FX9 shows a significant negative of MLT with the NAO (R=-0.41 and p-value < 0.03)".

19. Line 390. "at times reaching 2 degrees C". This number should be reported as a rate, in units of change in temperature per unit time. Please specify the time interval over which this estimate is computed.

Based on the previous comments we have modified Figure 6, which now shows the trends. The maximum observed trend with statistical significance is 0.08 C/year at SI8, which over roughly 30 years is about 2.4C. This station is considered a transition station and it is on the pathway of the progress of the alpha-ocean into the north. We have now modified the text using trends instead of absolute values. Once more, thanks for this observation.

**20. Minor grammatical points**

Line 17: "sections" -> "section"

Lines 19 and 20: It would be good to make a decision about consistent capitalization of regions ("South" or "south", etc.)

Line 24. "alternate the temperature and salinity contribution to stratification". Wording is unclear. Maybe the authors could write, "while in the North, the North Icelandic Irminger Current and East Icelandic Current alternate seasonally, shifting the region between temperature-dominated and salinity-dominated stratification."

Line 28. Remove comma after "locally".

Line 32. "their link" Wording is confusing. The word their implies a plural reference point, but the grammatical structure of the sentence does not clearly identify what this reference should be. Maybe "This study provides an unprecedented and detailed description of the seasonal to multi-decadal variability of mixed-layer depth and stratification around Iceland, showing links between this regional variability and the changing North Atlantic...." Lines 42-46. Capitalization and punctuation are inconsistent for numbered points. All three items could be capitalized as separate sentences, or all

three could be started with lower case letters, with semi-colons to separate the items. But mixed punctuation and capitalization is confusing.

Line 45. Remove "with"

Line 55. "drives" -> "drive"

Line 57. "heat fluxes are the main drivers" or "heat flux is the main driver"?

Line 57. "on the center" -> "in the center"

Line 58. "Nordic Seas have been previously described as a 'melting pot'". Inconsistent plurals. The Nordic Seas region is a melting pot? Or Nordic Seas are melting pots?

Line 59. "Nordic Seas are also a large repository". Same thing. "The Nordic Seas region is a large repository"?

Line 65. "of the Arctic Amplification" -> "of Arctic Amplification"

Line 65. "the decrease" -> "a decrease"

Line 74. "forcings" -> "forcing"

Line 74. "to control" -> "for controlling"

Line 74. Add comma after "mixing"

Line 76. "of the strong" -> "of strong"

Line 99. "hinders" -> "hinder"

Line 101. Add comma after "MLD"

Line 114. "IB" -> "IH"

Lines 128-137. It's standard practice to subscript theta in sigma theta.

Line 137. The line following equation (10) continues the sentence containing equation 1 and should not be capitalized or start a new paragraph. Equation 1 should be punctuated with a comma rather than a period

Line 137. "decomposed on" -> "decomposed to show"

Line 138. "contribution of the salinity" -> "contribution of salinity"

Line 146. "Where" is a continuation of the sentence containing equations (3) and (4). No capitalization and no indenting.

Line 148. Add comma after "salinity" since this is a compound sentence.

Line 151. "have" -> "has". (The sentence structure implies that only one component needs to have an impact, so the verb should assume a singular subject.)

Line 164. No indent. Please check all equations for this issue.

Line 187. "strike out" has a couple of distinct usages, but this reads as if it is using the baseball metaphor, which means to fail completely. Maybe use "are strikingly saltier".

Line 205. Missing words. Maybe "despite the fact that stratification ...."

Line 251. "correlate" -> "correlate"

Line 258. "in the order" -> "on the order" OR "are the same order of magnitude as"

Line 261. "neither .... nor" is not used correctly here. Change to "do not seem correlated with the MLT/MLS or with the ....:

Line 264. "it is". The text is not clear about the meaning of "it". Clarify whether "it" is station LB8 or the winter MLD.

Line 266. Start a new sentence: "front, fresher" -> "front. Fresher". Add a verb: "MLs associated" -> "MLs are associated"

Line 270. "of NAO" -> "of the NAO"

Line 271. "MLS" -> "MLS,"

Lines 325-326. "distribution (Fig. 8) ... southern (northern) ... alpha- (beta-) ..." Avoid using opposites in parentheses since opposites are also used for clarifications (e.g. Fig. 8 is likely not a match to "northern"). In general, this opposite-in-parentheses structure is difficult for readers to parse. If the point is worth making, then it can be spread into two sentences.

Line 325 and discussion of Fig. 8. The method underlying the results in Fig. 8 is shown in the Introduction. Here the text could reference Equation (5) to point readers to the relevant aspect of the computational approach.

Line 338. "200" -> "2000"?

200 is correct, we are trying to capture only the upper ocean.

Line 364. "northwestward, hence". This is a comma splice. Start a new sentence instead.

Line 366. Don't put opposites in parentheses. Write a clear two-part sentence instead.

Line 370. "variability, they". Comma splice. Start a new sentence with "they"

Line 374. Add a comma after "ML's"

---

## Author Comment (AC2)

**Reviewer 2**

egusphere-2025-2102

Stratification and mixed layer depth around Iceland, characterization and inter-annual variability

by A. Ruiz-Angulo, E. Portela, C. de Marez, A. Macrander, S. R. O´lafsdo´ttir, T. Meunier, S. Jo´nsson, and M. D. Pe´rez-Herna´ndez

In this manuscript, the rich hydrographic data set collected through long-term monitoring efforts is used to characterize stratification and mixed-layer properties around Iceland. Three distinct regions are identified, characterized as  $\alpha$ -,  $\beta$ -, or transition-oceans based on whether temperature or salinity dominates stratification, and implications for water mass transformation are discussed. Straddling the Greenland-Scotland Ridge, Iceland is situated at a key location for the overturning circulation in the Atlantic Ocean, particularly as regards the supply of dense water masses to its lower limb. I think this is an interesting and timely contribution to improve our understanding of the unique importance of Icelandic waters, but I have some concerns about the determination of mixed-layer depths, water mass definitions, and the one-dimensional mixed-layer model simulations that I hope the authors will address.

**Major comments:**

Your method for determining mixed-layer depths, with an adjusted density criterion, is commonly used. Verifying the outcome of the routine by visual inspection is a good way to ensure that the results are realistic. However, I think that you also need to document this, or at least be more specific about what the mixed layer depth you arrive at represents. Is it the surface mixed layer that is currently in direct contact with the atmosphere or the deepest weakly-stratified layer formed the same winter that may better represent the depth that convection has reached? In the subpolar North Atlantic and the Nordic Seas stacked mixed layers are prevalent and automatic routines are generally not very adept at identifying the depth of wintertime convection (Pickart *et al.*, 2002; Va ge *et al.*, 2015; Brakstad *et al.*, 2019). I encourage the authors to consider additional verification of the mixed-layer depths and expand the discussion about the limitations of the routine and what the outcome actually represents.

Indeed, estimates of MLD are or could be misleading. We understand the referee is particularly concerned about stacked mixed layers. For the summer profiles north of Iceland we do not see substantial evidence of remaining ML from the preceding winter. For the winter and particularly in the south of Iceland, the deep convection is so intense that we do not see any residual ML.

We have elaborated in the revised version of the manuscript about the validity or our method and that we are careful of not picking residual ML's. Taking advantage of your suggestion and also the references, we looked at the glider data from the Iceland Sea that we found in Våge et al., 2018. Since the data is open-access, we computed the MLD with our method (modified density threshold) to reproduce the MLD shown in that manuscript. Surprisingly, a quick check of the values gives a good qualitative agreement. The Iceland Sea is a challenging region as the winter ML can be shallow and quite complex in the internal structure. This structure is often compensated for in density and this is why we believe our method works.

We can confirm that our method with the density calculation and adapted threshold works quite well. Moreover, we corroborate as well that the temperature threshold method does not work in this region. We had come up with that conclusion before but now we have an extra verification bench mark for this.

More details about the PWP model (Price *et al.*, 1986) should be included in sections 2 (Data and methods) and 4 (MLD driving mechanisms from a 1D model). In particular, I miss information about how the model functions, about the initial conditions and forcing time series, and about the integration period and time step used for the simulations. I am sceptical about some of the results from the simulations, in particular the substantial impact of freshwater forcing on stations FX9 and ST5. Freshwater fluxes are usually of lesser importance for the mixed-layer development and tend to restratify the mixed layer (i.e., have a stabilizing influence) (e.g., Brakstad *et al.*, 2019), yet on these two stations they appear to cause a mixed-layer deepening of more than 300 m, which would require substantial evaporation in a region where precipitation generally dominates. At these two stations, wind-mixing is also surprisingly deep. Please confirm that the model simulates the mixed-layer development correctly and expand the discussion about these surprising results.

The information on the model inputs and details has been better detailed in Section 2 and supplementary material attending to this comment and a similar one from Reviewer 1. Likewise a new figure with the model outputs have been drafted for summer as it had older data, thanks for noticing it. We have also included the plots for the initial conditions and forcing time series in the supplementary materials.

I am not fully convinced that these water mass definitions (Figure 2, Table 2), dating back to Rudels *et al.* (2005) and primarily focused on polar conditions, have high fidelity around Iceland today. For example, the 0°C isotherm no longer provides accurate distinction between Atlanticand Arctic-origin water masses (Va°ge *et al.*, 2022). Perhaps these definitions should be updated or adapted for current conditions around Iceland? It matters when these definitions, without additional evidence, are used to determine the origins of different water masses. For example, are you confident that the dominant water masses near the surface at SI8 are of polar origin (line 172)? At times, the cold, fresh profiles in Figures 2c and 2d are unmistakably polar

(compare to Figures 11c and 12c in Rudels *et al.*, 2005), but while there may be some trace of polar water at SI8 the rest of the time I am not convinced that it is the predominant surface water mass. The same applies to the Atlantic-origin water at the same station (line 173). This could be more recently-formed intermediate water in the Iceland Sea that may be warmer than 0°C (Va°ge *et al.*, 2022). A closer inspection of each profile, to verify that it displays the intermediate temperature- and salinity maxima characteristic of Atlantic-origin water (Mauritzen, 1996), would likely be necessary for unambiguous identification.

It is a valid point, we understand the concern of the reviewer about the definitions of water masses; we looked at the suggested reference (Våge et al., 2022) and we cannot find any definition that could be used to correct ours. We did look at Swift and Aagaard, 1981, who suggested a similar cautionary message about the 0 C limit. We have updated our manuscript by adding the full reference to the definitions used in Casanova et al., 2020, which are based on definitions from Rudels et al., 2005 and Våge at al., 2011. We would be very happy to re-define the water masses if the reviewer could kindly point at us what would be the best definition we have for Icelandic waters. Recent studies support this simple definition using radioactive tracers, those techniques seem to clearly distinguish PSW and on a TS space they do fall in the same definition we are using (Dale et al., 2024). While we understand these definitions should be carefully used or revisited, this is not the main goal of our paper. The main objective of our manuscript is to highlight the richness of watermasses around Iceland abd we used Casanova-MAsjoan et al. (2020) to be consistent with previous studies done with the dataset (e.g. Centyfoyen et al. 2023).

Particularly, for SI8 we have discussed with our local experts whol just confirmed in SI8 CTD data from 2005 – 2019: Water of polar origin is present at the surface in most profiles at SI8, as indicated by salinity <34.9. While in summer, there is a rather shallow (<100 m) but very fresh PSW layer above a subsurface AW core, in winter the ML reaches deeper (150 m) and salinity is higher, likely due to some AW contribution. This is the case both in winter (upper 150 m) and in summer (less deep, but even lower salinities). Typically, ML salinity at SI8 is even lower than 34.8, so the exact criterion does not change this notion.

In deeper layers, a gradual warming is observed as elsewhere in the deep Nordic Seas. Using the 0°C criterion to distinguish between ArOW and AtOW results in a reduced fraction of ArOW which might be biased. However, only the deepest and densest part of the profiles falls into the ArOW category. The water formed in the winter ML at SI8 was below 0°C until ca. 2000, but predominantly >0°C since then. This would indeed change the classification from ArOW to AtOW although the physics remained the same, only at a higher temperature level.

Nevertheless, our study is not based on water mass transformations thus we have added a caveat on this concern stating that the fixed definitions might lead to biased estimates as waters are generally warmer than 20 years ago.

In the revised version we have added the following sentence: "It is noteworthy that using fixed definitions of water masses may lead to biased estimates, as these water masses have been steadily warming over the past two decades".

**General comments:**

The paragraph on Atlantification of the Arctic Ocean in the introduction (line 63) appears unmotivated and its purpose is not immediately clear. The transition into an  $\alpha$ -ocean north of Iceland is later ascribed to Atlantification, though Atlantification is more generally considered where Atlantic properties displace Polar properties in the Arctic Ocean. If applicable north of Iceland, I think you need to start making a stronger case for the relevance of Atlantification already in the introduction and follow that up in the discussion (e.g., line 373).

This is a great observation, often Atlantification is associated with the Barrents, Kara and Eurasian basins. We conceptualized Atlantification as an increasing influence of Atlantic Water (AW) in the Arctic Ocean, which has consequences such as the ones we observe. For example warmer subsurface layers including the mixed layer. Iceland is at a unique boundary, a sub-Arctic region that shares both Arctic and Atlantic characteristics. I believe that questioning these concepts in the light of the alpha-ocean propagation north of Iceland would be valid, and may represent a process that has not been reported before within the Atlantification umbrella.

Why did you limit the data set to include observations only to 2019 (line 105)? The MFRI monitoring program is ongoing, so more recent data should be available. You document interesting signals toward the end of the record (Figure 5) that could be further examined.

This is a very good point, the datasets from MFRI are/were not open access. We used the data that we had access to. The current datasets are also being post-processed and quality-controlled. Perhaps we could have added one or two more years but we do not think this would have made any relevant change to our analysis. Finally, we would have loved to have access to all the possible databases as we believe in open-access policies and we know this is currently changing within the Institute.

Did you consider including other stations than the deepest on each transect in the analysis? At least some of the transects have several stations offshore of the shelf break. Perhaps for one transect in each hydrographic region, it would be very interesting to know if the other deep stations corroborate the results – that would make your conclusions more robust.

This is a very good point, we did try to increase the number of stations for our study but this was resulting in a larger set of results that we found harder to pack together in a single manuscript. The variability resulting from using more stations increased, thus, we decided to only carry on this analysis for only the deepest stations.

While LB8 may be representative for a β-ocean regime, located in the middle of Denmark Strait it is also characterized by substantial high-frequency variability (von Appen *et al.*, 2014; Lin *et*

al., 2020). Perhaps KG6 would be a better choice for investigating interannual to decadal variability in mixed-layer properties in this region?

Following the reviewer's suggestion we have tried and replaced LB station by KG in Figure 5. (see below). However, KG is less well sampled, and some of the years are missing. Moreover, even if it is true that LB station might be subjected to higher high-frequency variability, this variability would be filtered out by the annual means. Since Figure 6 already shows all the stations, we think that keeping LB is a better choice for figure 5.

I think the text would be more clear if the past tense was consistently used for past events (e.g., line 248). At least, avoid switching between past and present tenses within the same paragraph.

Thanks for the suggestion, we have tried to correct this and hopefully we did not miss this suggestion throughout the text.

The NAO primarily represents atmospheric variability on time scales shorter than the ocean may fully adjust. Variability in circulation and mixed-layer depth/properties may be expected (e.g., Yashayaev, 2007), but to a lesser extent wholesale changes throughout the water column, hence line 257 could be phrased more conservatively.

We agree with the referee about this lag and we have now added a paragraph with the correlation values and the lags. The paragraph now reads: "The observed variability of the ML and its temperature in FX9 exhibits certain correlation (R) with the NAO; for MLD, the best correlation was R=0.53, p-value<0.01 at lag zero; for MLS R=-0.52, p-value<0.01 at lag -2 years (NAO leading), and for MLT R=-0.49, p-value<0.01 at lag -1 year (NAO leading (Fig. 5g, h). However, we consider that a 2-year lag lacks a realistic physical explanation, thus, we prefer to not to consider this as a reliable correlation. More qualitatively, positive NAO at the beginning and the end of the time series, corresponds with deeper colder and fresher MLs, while negative NAO between 2000 and 2015 roughly corresponds with shallower, warmer, and saltier MLs".

If the mean mixed-layer depth at SI6 is less than 100 m (Figure 4), how can anomalies in mixed-layer depth exceed 100 m in both positive and negative directions (Figure 5b, e)?

This is because mixed layer depth anomalies are expressed in percentage. We chose this metric because it takes into account the variability regarding the MLD value, as, let's say, a 20-m anomaly is not as important in a mixed layer of 50 m as in a mixed layer of 300 m depth.

For the statistical analyses (Figures 5 and 6), please provide the values for the correlations and trends reported in the text as well as their uncertainties. That would help make the analysis more quantitative and robust.

Thank you for your suggestion. We have now computed Pearson correlations between all variables and the results are detailed in Table 1.

Table 1: Pearson correlations between different variables. Non significant correlations have been omitted. The shown correlations are significant at 95% confidence (p<0.05), and those significant at 99% (p<0.01) are in bold.

|         | FX          |         | SI          |         | LB          |         |
|---------|-------------|---------|-------------|---------|-------------|---------|
|         | Correlation | p-value | Correlation | p-value | Correlation | p-value |
| MLT-MLS | R=0.69      | p<0.01  | R=0.74      | P=<0.01 | R=0.95      | P<0.01  |
| MLT-MLD | -           | -       | -           | -       | R=0.76      | P<0.01  |
| MLS-MLD | -           | -       | -           | -       | R=0.68      | P=<0.01 |
| NAO-MLD | R=0.53      | P<0.01  | -           | -       | -           | -       |
| NAO-MLT | R=-0.41     | P<0.03  | -           | -       | -           | _       |
| NAO-MLS | -           | -       | -           | -       | -           | -       |

We have also computed lagged cross correlations between the mixed layer properties and the NAO index.

Please provide some information about the gridding or averaging used for the data set that you calculate the spice frequency shown in Figure 8 from.

Thanks for pointing out the missing information. Figure 8 uses the Brakstad et al (2023) hydrographic dataset, which is an open source product largely used for studies in the Arctic and SubArctic. We have made reference to this dataset at the end of the section Data and Methods. The gridding is explained in the reference and the method used for this product derivation. The spice frequency is described in Strehl et al. (2024).

**Detailed comments:**

**Line 16:**

Dense-water formation is generally considered a necessary condition for a global overturning circulation rather than a driving mechanism (e.g., Kuhlbrodt *et al.*, 2007).

**Line 24:**

Contributions should be in plural. Fixed

**Line 40:**

As a proper name, ridge must be capitalized in the Greenland-Iceland-Scotland Ridge. Fixed

**Line 44:**

The Faroe Islands would be more appropriate than just Faroe. Fixed

**Line 52:**

Regarding the transformation of AW into AtOW, it is unclear what is meant by "along the Norwegian Current western intrusions". Presumably that water mass transformation occurs along the Norwegian Current, which is a main mode of water mass transformation that is otherwise not mentioned here (e.g., Mauritzen, 1996; Huang *et al.*, 2023). Thanks for the observation, we have now removed the "western intrusions" to keep the text simpler.

**Line 56:**

Very likely heat loss is important also for water mass transformation east of Greenland, but wind forcing and sea-ice retreat are necessary for preconditioning this region before convection can occur. We have tried to re-phrase this sentence but we are still unsure about the specific change the reviewer is suggesting. Wewould appreciate a more detailed recommendation to address this point.

**Lines 60:**

Do you mean from the Arctic Ocean? The Arctic, by itself, is ill-defined and has multiple interpretations. The reference to Arctic as north and northwest of Iceland (line 91) does not help clarifying. Indeed it is an ill-defined, we believe in this case it is the Arctic as the main source of the freshwater is on land and would not necessarily mean Arctic Ocean. Nevertheless the one in line 91 we completely agree that should be Arctic Ocean, thanks for pointing that out.

**Line 64:**

The word "extent" is used twice in the same sentence. Thanks, fixed.

**Line 71:**

Please be more specific, under what conditions does wind forcing enhance turbulent mixing that deepens the mixed layer? The following sentence does not add clarification regarding the impact of wind forcing, even though it alludes to an example of this process.

**Line 79:**

The definite article in front of 40% should be removed. Thanks, fixed.

**Line 82:**

It would be good to be more specific about what kind of mixing you refer to, perhaps diapycnal mixing or convection? Thanks, fixed.

**Line 170:**

There should be a comma after southwest of Iceland. Thanks, fixed.

**Line 185:**

A reference to Jo'nsson and Valdimarsson (2012) would be appropriate here. Thanks, fixed.

**Line 214:**

What do you mean by "hydrographic onset"? We have rephrase this to hydrographic characteristics

**Line 226:**

The sentence starting with "Therefore, ..." is unclear and should be rephrased. Thanks, fixed.

**Line 228:**

It should be: "... the potential for deep convection ..." Thanks, fixed.

**Line 241:**

The paragraph starting on this line is repetitive and could be written more concisely. Thanks, fixed.

**Line 256:**

The Feucher et al. (2022) paper is missing from the reference list. Thanks, fixed.

**Line 278:**

Please specify what the anomalies are computed from. Are they anomalies relative to the record-long mean at each station? Yes, they are relative to the record long. We have now added this to the manuscript.

**Line 286:**

The term aliasing has a specific meaning in time series analysis, where high-frequency variability appears as a lower-frequency signal due to inadequate sampling. It is not clear that the term was used appropriately in this sentence.

We agree with the reviewer, aliasing is not a suitable word here and it has been modified in the revised manuscript.

**Line 289:**

The reference to Polyakov *et al.* (2017) is not appropriate here, they did not consider the mixed layers around Iceland or the NIIC in their study. Thanks, removed.

**Line 295:**

Mixed layers in summer are shallow primarily because the buoyancy forcing abates in spring and becomes largely positive in summer, episodic wind and wave mixing (also reduced in summer) ensure that there is a mixed layer. This is a main reason why stratification is higher in summer than in winter. Please rephrase. Thanks for the complementary information. In this sentence we are not discussing the mechanism that sustains the ML here; rather, we are stating the observation that summer stratification is high and that he MLD is shallow. We believe that adding a discussion of the underlying mechanism at this point would be confusing.

**Line 311:**

Do you mean SI8 rather than LB8, if you refer to the region north of Iceland? We started the discussion with LB8 and we considered the stations KG6, SI8 northeast, which are b,c,d. We have now explicitly written those names in the text.

**Line 312:**

Wind-stress, by itself, will not cause convection - for that you need buoyancy forcing. This process would more appropriately be referred to as wind-mixing.

We agree with the reviewer and we have changed this sentence. We were trying to point out that station LB is equally balanced with wind mixing and convection. We tried to make the difference that it has some convection potential from the AW but it is reduced compared to the southern stations that have only AW. We have now added "wind-mixing".

**Line 350:**

There is at least a preposition missing near the end of this sentence. Thanks, fixed.

**Line 356:**

The surface water masses of the EGC would have a strong Polar (not Arctic) influence. Thanks, fixed.

**Line 375:**

An increasingly temperature-dominated stratification may in principle lead to deeper convection, but reduced atmospheric forcing and shallower, less dense mixed layers (Moore *et al.*, 2015; Va ge *et al.*, 2022) strongly suggest that the Iceland Sea is not likely to become an important source to the NIJ. Indeed, we agree with the reviewer, we are not suggesting that this region would be an important source but more like it could contribute or even modify (by dyapicnal mixing) the dense flow carried by the NIJ.

**Line 390:**

The warming trend of 2°C, was that over the entire record length? When reporting a trend, please also provide a time reference. We are sorry for misleading, we are trying to report the mixed layer temperature range from winter to summer, not a warming trend. The warming trends are now reported in Figure 6.

Figure 5: It would have been illuminating to see panels of mixed-layer density anomalies as well. Thanks, Figure 5 now has been re-designed.

Figure 6:
I think it would be advantageous to use the same color scale for all panels, even if that means that some colors occasionally will saturate, to facilitate direct comparison between stations.

We have now re-designed this figure also from the comments of the other reviewer.

Figure 6, caption:

It should be: "... all stations except LB8 and KG6..."

Thanks, but now Figure 6 has been re-designed so we are no longer using variable colorbars.

**Figure 7:**

Please also show the total mixed-layer depth resulting from the combination of all driving mechanisms. Do they add linearly? While the observations stem from February and not end of winter when the mixed-layers are deepest, it appears that some of the simulated MLDs will be substantially deeper. In the text, individual panels are referred to by letters, but these letters are not visible on the figure. We think that the purpose of using this 1D model is to ilustrate the main mechanisms rather than to reproduce or analyze the full evolution of the MLD. We have indicated the observed MLD with a black dot in the figure representing the winter and the summer. Therefore, we would prefer to keep the figure as it is.

**Table 1:**

KG6 and HB6 are not located in the Greenland Sea. Referring to this region as Blosseville Basin or western Iceland Sea would be more appropriate. Thanks, fixed.

**Table 2:**

Some of the less than/greater than signs are in the wrong direction and contradict Figure 2 (e.g., SW temperature, warm PSW density, and AtOW density). Thanks a lot for this correction, now it is fixed

**References**

Brakstad A, Va ge K, Ha vik L, Moore GWK. 2019. Water mass transformation in the Greenland Sea during the period 1986-2016. *Journal of Physical Oceanography* 49: 121–140, doi:10.1175/JPO-D-17-0273.1.

Huang J, Pickart RS, Chen Z, Huang RX. 2023. Role of air-sea heat flux on the transformation of Atlantic Water encircling the Nordic Seas. *Nature Communications* 14: doi:10.1038/s41 467–023–35 889–3.

Jo'nsson S, Valdimarsson H. 2012. Water mass transport variability to the north Icelandic shelf, 1994-2010. *ICES Journal of Marine Science*: doi:10.1093/icesjms/fss024.

Kuhlbrodt T, Griesel A, Montoya M, Levermann A, Hofmann M, Rahmstorf S. 2007. On the driving processes of the Atlantic Meridional Overturning Circulation. *Reviews of Geophysics* 45: RG2001, doi:10.1029/2004RG000 166.

Lin P, Pickart RS, Jochumsen K, Moore GWK, Valdimarsson H, Fristedt T, Pratt LJ. 2020. "kinematic structure and dynamics of the denmark strait overflow from ship-based observations". *Journal of Physical Oceanography* 50: 3235–3251, doi:10.1175/JPO–D–20–0095.1.

Mauritzen C. 1996. Production of dense overflow waters feeding the North Atlantic across the Greenland- Scotland Ridge. Part 1: Evidence for a revised circulation scheme. *Deep Sea Research I* 43: 769–806, doi:10.1016/0967–0637(96)00 037–4.

Moore GWK, Va ge K, Pickart RS, Renfrew IA. 2015. Decreasing intensity of open-ocean convection in the Greenland and Iceland Seas. *Nature Climate Change* 5: doi:10.1038/nclimate2688.

Pickart RS, Torres DJ, Clarke RA. 2002. Hydrography of the Labrador Sea during active convection. *Journal of Physical Oceanography* 32: 428–457.

Polyakov IV, Pnyushkov AY, Alkire MB, Ashik IM, Baumann TM, Carmack EC, Goszczko I, Guthrie J, Ivanov VV, Kanzow T, Krishfield R, Kwok R, Sundfjord A, Morison J, Rember R, Yulin A. 2017. Greater role for Atlantic inflows on sea-ice loss in the Eurasian Basin of the Arctic Ocean. *Science* 356: 285–291, doi:10.1126/science.aai8204.

Price JF, Weller RA, Pinkel R. 1986. Diurnal cycling: Observations and models of the upper ocean response to diurnal heating, cooling, and wind mixing. *Journal of Geophysical Research* 91: 8411–8427, doi:10.1029/JC091iC07p08 411.

Rudels B, Bjo rk G, Nilsson J, Winsor P, Lake I, Nohr C. 2005. The interaction between waters from the Arctic Ocean and the Nordic Seas north of Fram Strait and along the East Greenland Current: Results from the Arctic Ocean-02 Oden expedition. *Journal of Marine Systems* 55: 1–30.

Va ge K, Moore GWK, Valdimarsson H, Jo nsson S. 2015. Water mass transformation in the Iceland Sea. *Deep Sea Research I* 101: 98–109, doi:10.1016/j.dsr.2015.04.001.

Va ge K, Semper S, Valdimarsson H, Jo nsson S, Pickart RS, Moore G. 2022. Water mass transformation in the Iceland Sea: Contrasting two winters separated by four decades. *Deep Sea Research I*: doi:10.1016/j.dsr.2022.103 824.

von Appen WJ, Pickart RS, Brink KH, Haine TWN. 2014. Water column structure and statistics of Denmark Strait Overflow Water cyclones. *Deep Sea Research I* 84: 110–126, doi:10.1016/j.dsr.2013.10.007.

Yashayaev I. 2007. Hydrographic changes in the Labrador Sea, 1960-2005. *Progress in Oceanography* 73: 242–276, doi:10.1016/j.pocean.2007.04.015.